# Individual Privacy Accounting via a Rényi Filter

**Vitaly Feldman**
Apple
vitaly.edu@gmail.com

**Tijana Zrnic**
University of California, Berkeley
tijana.zrnic@berkeley.edu

## Abstract

We consider a sequential setting in which a single dataset of individuals is used to perform adaptively-chosen analyses, while ensuring that the differential privacy loss of each participant does not exceed a pre-specified privacy budget. The standard approach to this problem relies on bounding a worst-case estimate of the privacy loss over all individuals and all possible values of their data, for every single analysis. Yet, in many scenarios this approach is overly conservative, especially for "typical" data points which incur little privacy loss by participation in most of the analyses. In this work, we give a method for tighter privacy loss accounting based on the value of a personalized privacy loss estimate for each individual in each analysis. To implement the accounting method we design a *filter* for Rényi differential privacy. A filter is a tool that ensures that the privacy parameter of a composed sequence of algorithms with *adaptively-chosen* privacy parameters does not exceed a pre-specified budget. Our filter is simpler and tighter than the known filter for $(\epsilon, \delta)$-differential privacy by Rogers et al. [29]. We apply our results to the analysis of noisy gradient descent and show that personalized accounting can be practical, easy to implement, and can only make the privacy-utility tradeoff tighter.

## 1 Introduction

Understanding how privacy of an individual degrades as the number of analyses using their data grows is of paramount importance in privacy-preserving data analysis. This allows individuals to participate in multiple disjoint statistical analyses, all the while knowing that their privacy cannot be compromised by aggregating the resulting reports. Furthermore, this feature is crucial for privacy-preserving algorithm design—instead of having to reason about the privacy properties of a complex algorithm, it allows reasoning about the privacy of the subroutines that make up the final algorithm.

For differential privacy (DP) [13], this accounting of privacy losses is typically done using composition theorems. Importantly, given that statistical analyses often rely on the outputs of previous analyses, and that algorithmic subroutines feed into one another, the composition theorems need to be *adaptive*, namely, allow the choice of which algorithm to run next to depend on the outputs of all previous computations. For example, in gradient descent, the computation of the gradient depends on the value of the current iterate, which itself is the output of the previous steps of the algorithm.

Given the central role of adaptive composition theorems in differential privacy, they have been investigated in numerous works (e.g. [16, 23, 11, 27, 26, 4, 29, 8, 30]). While they differ in some aspects, they also share one limitation. Namely, all of these theorems reason about the worst-case privacy loss for each constituent algorithm in the composition. Here, "worst-case" refers to the worst choice of individual in the dataset and worst choice of value for their data. This pessimistic accounting implies that every algorithm is summarized via a single privacy parameter, shared among all participants in the analysis.

In most scenarios, however, different individuals have different effects on each of the algorithms, as measured by differential privacy. More precisely, the output of an analysis may have little to no

35th Conference on Neural Information Processing Systems (NeurIPS 2021).

dependence on the presence of some individuals. For example, if we wish to report the average income in a neighborhood, removing an individual whose income is close to the average has virtually no impact on the final report after noise addition. Similarly, when training a machine learning model via gradient descent, the norm of the gradient given by a data point is often much smaller than the maximum norm (typically determined by a clipping operation). As a result, in many cases no single individual is likely to have the worst-case effect on all the steps of the analysis. This means that accounting based on existing composition theorems may be unnecessarily conservative.

In this work, we present a tighter analysis of privacy loss composition by computing the associated divergences at an individual level. In particular, to achieve a pre-specified privacy budget, we keep track of a personalized estimate of the privacy loss divergence for each individual in the analyzed dataset, and ensure that the respective estimate is maintained under the budget for all individuals throughout the composition. We do so by applying each analysis only to the points that are estimated to have sufficient leftover privacy budget.

## 1.1 Overview of main results

It is feasible to measure the worst-case effect of a fixed data point on a given analysis in terms of DP. One can simply replace the supremum over all datasets in the standard definition of (removal) DP with the supremum over datasets that include that specific data point (see Definition 2.5). However, a meaningful application of adaptive composition with such a definition immediately runs into a technical challenge: standard composition theorems require that the privacy parameter of each step be fixed in advance. For individual privacy, this approach requires using the worst-case value of the individual privacy loss over all the possible analyses at a given step. Individual privacy parameters are much more sensitive to the choice of analysis than worst-case parameters, and thus maximizing over all analyses is likely to negate the benefits of using individual losses in the first place.

Thus the main technical challenge in analyzing composition of individual privacy losses is that they are themselves random variables that depend on the outputs of all previous computations. If we denote by $a_1, \ldots, a_t$ the output of the first $t$ adaptively composed algorithms $\mathcal{A}_1, \ldots, \mathcal{A}_t$, then the individual privacy loss of any point incurred by applying algorithm $\mathcal{A}_{t+1}$ is a function of $a_1, \ldots, a_t$. Hence, to tackle the problem of composing individual privacy losses we need to understand composition with *adaptively-chosen* privacy parameters. We refer to this kind of composition as *fully adaptive*.

The setting of fully adaptive privacy composition is rather subtle and even defining privacy in terms of the adaptively-chosen privacy parameters requires some care. This setting was first studied by Rogers et al. [29], who introduced the notion of a *privacy filter*. Informally, a privacy filter is a stopping time rule that halts a computation based on the adaptive sequence of privacy parameters and ensures that a pre-specified privacy budget is not exceeded. Rogers et al. define a filter for approximate DP that asymptotically behaves like the advanced composition theorem [16], but is substantially more involved and loses a constant factor. Moreover, several of the tighter analyses of Gaussian noise addition require composition to be done in Rényi differential privacy (RDP) [1, 26].

Our main result can be seen as a privacy filter for RDP which justifies stopping the analyses based on the sum of privacy parameters so far *even* under fully adaptive composition.

**Theorem 1.1.** *Fix $B \geq 0, \alpha \geq 1$. Assume $\mathcal{A}_t$ is $(\alpha, \rho_t)$-RDP, where $\rho_t$ is any function of $a_1, \ldots, a_{t-1}$. If $\sum_{t=1}^{k} \rho_t \leq B$ holds almost surely, then the adaptive composition of $\mathcal{A}_1, \ldots, \mathcal{A}_k$ is $(\alpha, B)$-RDP.*

When $\rho_1, \ldots, \rho_k$ are fixed, Theorem 1.1 recovers the usual composition result for RDP [26]. Our RDP filter immediately implies a simple filter for approximate DP that is as tight as any version of the advanced composition theorem obtained via concentrated DP [4]. These Rényi-divergence-based analyses are known to improve on the classical rate [16] and, in particular, improve on the rate in [29].

We instantiate our general result for fully adaptive composition in the setting of individual privacy accounting. This allows us to define an *individual privacy filter*, which, given a fixed privacy budget, adaptively drops points from the analysis once their *personalized* privacy loss estimate exceeds the budget. Therefore, instead of keeping track of a single running privacy loss estimate for all individuals, we track a less conservative, personalized estimate for each individual in the dataset. Individual privacy filtering allows for better, adaptive utilization of data points for a given budget. It can also naturally be applied to accounting in the local DP model, whereby each user stops responding once their local implementation of the filter indicates that their personal privacy budget is exhausted.

Individual privacy parameters are particularly easy to compute for linear queries, as well as their high-dimensional generalizations. We show that our technique gives an algorithm for answering a sequence of adaptively-chosen linear queries that are sparse across time, meaning that, for any user, the number of queries that are non-zero on that user's data is small. Such queries arise, for example, when a platform counts the number of users that participate in certain activities (the type of activity being adaptive to the data collected in the previous days) and users generally participate in a small number of activities. Formally, a special case of our result implies the following theorem.

**Theorem 1.2.** *There exists an algorithm $\mathcal{A}$ that, given a dataset $S = (X_1, \ldots, X_n) \in \mathcal{X}^n$, sparsity parameter $s$ and privacy level $\kappa$, for any adaptively-chosen sequence of queries $q_1, \ldots, q_k$ of arbitrary length $k$, where $q_i \colon \mathcal{X} \to \{0, 1\}$, provides a sequence of answers $a_1, \ldots, a_k$ such that:* (1) *$\mathcal{A}$ is $(\alpha, \alpha\kappa)$-RDP for all $\alpha \geq 1$;* (2) *for all $t$ and any $\delta \in (0, 1)$, the probability that $|a_t - \sum_{X_i \in S_t} q_t(X_i)| > \sqrt{s \log(1/\delta)/\kappa}$ is at most $\delta$, where $S_t = (X_i \in S : \sum_{j=1}^{t} q_j(X_i) \leq s)$.*

We note that the provided answers are guaranteed to be accurate only as long as the queries are truly sparse, meaning $\sum_{j=1}^{t} q_j(X_i) \leq s$ for (almost) all $i \in [n]$. This follows because the queries are accurate on the set $S_t$, hence $S_t$ needs to be similar to $S$ for the queries to be accurate on $S$. The privacy guarantee, on the other hand, holds for *any* sequence of queries of *any* length $k$. We describe a more general version of this result in Section 4.2. A natural application of our general theorem is the setting of high-dimensional linear queries generated by gradient descent. We apply our theory to the analysis of private gradient descent [1], and show—both theoretically and empirically—that individual accounting can be easy to implement and can only make the resulting privacy-utility tradeoff tighter. Independently, without any individual accounting, in our empirical evaluations we also observe that private batch gradient descent, when tuned appropriately, outperforms private stochastic gradient descent in terms of the privacy-utility tradeoff. While we make this observation only on MNIST, we believe this phenomenon holds more generally and is worth further investigation.

## 1.2 Related work

The main motivation behind our work is obtaining tighter privacy accounting methods through, broadly speaking, "personalized" accounting of privacy losses. Existing literature in DP discusses several related notions [22, 19, 31, 6], although typically with an incomparable objective. Ghosh and Roth [22] discuss individual privacy in the context of selling privacy at auction and their definition does not depend on the value of the data point but only on its index in the dataset. Cummings and Durfee [6] rely on a similar privacy definition, investigate an associated definition of individual sensitivity, and demonstrate a general way to preprocess an arbitrary function of a dataset into a function that has the desired bounds on individual sensitivities.

Ebadi et al. [19] introduce personalized DP in the context of private database queries and describe a system which drops points when their personalized privacy loss exceeds a budget. While this type of accounting is similar to ours in spirit, their work only considers basic and non-adaptive composition. Wang [31] considers the privacy loss of a specific data point relative to a fixed dataset and provides techniques for evaluating this "per-instance" privacy loss for several statistical problems. Wang also briefly discusses adaptive composition of per-instance DP as a straightforward generalization of the usual advanced composition theorem [16], but the per-instance privacy parameters are assumed to be *fixed*. As discussed above, having fixed per-instance privacy parameters, while allowing adaptive composition, is likely to negate the benefits of personalized privacy estimates. The work of Ligett et al. [25] tightens individuals' personalized privacy loss by taking into account subsets of analyses in which an individual does not participate. Our work naturally captures this setting while allowing full adaptivity. Moreover, they consider the usual worst-case privacy loss, rather than an individual one, and the analyses in which a user participates are determined in a data-independent way.

Our work can be seen as related to data-dependent approaches to analyses of privacy-preserving algorithms such as smooth sensitivity [28], the propose-test-release framework [9], and *ex-post* privacy guarantees [32]. Our results are complementary in that we aim to capture the dependence of the output on the value of each individual's data point as opposed to the "easiness" of the entire dataset. Our approach also relies on composition to exploit the gains from individual privacy accounting.

Finally, adaptive composition of DP is a key tool for establishing statistical validity of an adaptively-chosen sequence of analyses [18, 17, 3]. In this context, Feldman and Steinke [21] show that individual KL-divergence losses (or RDP losses for $\alpha = 1$) compose adaptively and can be used to

derive tighter generalization results. However, this result still requires that the average of individual KL-divergences be upper bounded by a fixed worst-case value and the analysis is limited to $\alpha = 1$.

## 2 Preliminaries

We let $S = (X_1, \dots, X_n)$ denote the analyzed dataset, and $S^{-i} \stackrel{\text{def}}{=} (X_1, \dots, X_{i-1}, X_{i+1}, \dots, X_n)$ the analyzed dataset after removing point $X_i$. We will generally focus on algorithms that can take as input a dataset of arbitrary size. If, instead, the algorithm requires an input of fixed size, one can obtain the same results for algorithms that replace $X_i$ with an arbitrary fixed element $X^\star$ (e.g. 0).

**Definition 2.1** ([13, 12]). *A randomized algorithm $\mathcal{A}$ is $(\epsilon, \delta)$-differentially private (DP) if for all datasets $S = (X_1, \dots, X_n)$, $i \in [n]$, and measurable sets $E$,*

$$\Pr\left[\mathcal{A}(S) \in E\right] \leq e^\epsilon \Pr\left[\mathcal{A}(S^{-i}) \in E\right] + \delta, \text{ and } \Pr\left[\mathcal{A}(S^{-i}) \in E\right] \leq e^\epsilon \Pr\left[\mathcal{A}(S) \in E\right] + \delta.$$

Our analysis will rely on Rényi differential privacy (RDP), a relaxation of DP based on Rényi divergences which often leads to tighter privacy bounds than analyzing DP directly. Formally, the Rényi divergence of order $\alpha \in (1, \infty)$ between two measures $\mu$ and $\nu$ such that $\mu \ll \nu$ is defined as:

$$D_\alpha(\mu\|\nu) = \frac{1}{\alpha - 1} \log \int \left(\frac{d\mu}{d\nu}\right)^\alpha d\nu.$$

The Rényi divergence of order $\alpha = 1$ is defined by continuity, and recovers the KL-divergence. Relying on a common abuse of notation, we use $\mathcal{A}(\cdot)$ to refer to the output distribution of a randomized algorithm. Thus, $D_\alpha(\mathcal{A}(S)\|\mathcal{A}(S^{-i}))$ denotes the divergence between the output distribution of $\mathcal{A}$ on input $S$ and the output distribution of $\mathcal{A}$ on input $S^{-i}$. Similarly, we use $a \sim \mathcal{A}(S)$ to denote $a$ being sampled randomly from the output distribution of $\mathcal{A}$ on $S$. We also use $D_\alpha^{\leftrightarrow}(\mu\|\nu) \stackrel{\text{def}}{=} \max\{D_\alpha(\mu\|\nu), D_\alpha(\nu\|\mu)\}$ to denote the maximum of the two directions of Rényi divergence.

**Definition 2.2** ([26]). *A randomized algorithm $\mathcal{A}$ is $(\alpha, \rho)$-Rényi differentially private (RDP) if for all datasets $S = (X_1, \dots, X_n)$ and $i \in [n]$, $D_\alpha^{\leftrightarrow}\left(\mathcal{A}(S)\|\mathcal{A}(S^{-i})\right) \leq \rho$.*

A related notion that we will make use of is zero-concentrated differential privacy (zCDP).

**Definition 2.3** ([4]). *A randomized algorithm $\mathcal{A}$ satisfies $\kappa$-zero-concentrated differential privacy (zCDP) if it satisfies $(\alpha, \alpha\kappa)$-RDP for all $\alpha \geq 1$.*

Although our guarantees will be stated in terms of RDP, the conversion to DP is immediate.

**Fact 2.4** ([26]). *If $\mathcal{A}$ is $(\alpha, \rho)$-RDP, then it is also $(\rho + \log(1/\delta)/(\alpha - 1), \delta)$-DP, for any $\delta \in (0, 1)$.*

Our main object of study is *adaptive composition*. Here, for a given input dataset $S$ and a sequence of algorithms $(\mathcal{A}_t)_{t=1}^k$, one sequentially computes reports $a_t = \mathcal{A}_t(a_1, \dots, a_{t-1}, S)$ as a function of the previous reports and the input dataset. We denote by $a^{(t)} \stackrel{\text{def}}{=} (a_1, \dots, a_t)$ the sequence of the first $t$ reports, and by $\mathcal{A}^{(t)}(\cdot) \stackrel{\text{def}}{=} (\mathcal{A}_1(\cdot), \mathcal{A}_2(\mathcal{A}_1(\cdot), \cdot), \dots, \mathcal{A}_t(\mathcal{A}_1(\cdot), \dots, \cdot))$ the composed algorithm which produces $a^{(t)}$. If $\mathcal{A}_t(a_1, \dots, a_{t-1}, \cdot)$ is $(\alpha, \rho_t)$-RDP for *all* values of $a_1, \dots, a_{t-1}$, then the standard adaptive composition theorem for RDP says that $\mathcal{A}^{(k)}$ is $(\alpha, \sum_{t=1}^k \rho_t)$-RDP [26]. This implicitly assumes that $\rho_1, \dots, \rho_k$ are fixed and thus independent of the realized reports $a_1, \dots, a_k$.

**Individual privacy.** Our individual accounting relies on measuring the maximum possible effect of an individual data point on a dataset statistic in terms of Rényi divergence. This measure is equivalent to an RDP version of personalized DP [19]. For convenience we will refer to it as individual RDP. We note, however, that, by itself, a bound on this divergence does *not* imply any formal privacy guarantee for an individual, since the individual RDP parameter depends on the sensitive value of the data point.

**Definition 2.5** (Individual RDP). *Fix $n \in \mathbb{N}$ and a data point $X$. We say that a randomized algorithm $\mathcal{A}$ satisfies $(\alpha, \rho)$-individual Rényi differential privacy for $X$ if for all datasets $S = (X_1, \dots, X_m)$ such that $m \leq n$ and $X_i = X$ for some $i$, it holds that $D_\alpha^{\leftrightarrow}\left(\mathcal{A}(S)\|\mathcal{A}(S^{-i})\right) \leq \rho$.*

Therefore, to satisfy RDP, an algorithm needs to satisfy individual RDP for all data points $X$.

We give two simple examples of individual RDP computation.

**Example 2.6** (Linear queries). *Let $S = (X_1, \ldots, X_n) \in \mathcal{X}^n$. Suppose that $\mathcal{A}$ is a $d$-dimensional linear query with Gaussian noise addition, $\mathcal{A}(S) = \sum_{j \in [n]} q(X_j) + \xi$, for some $q \colon \mathcal{X} \to \mathbb{R}^d$ and $\xi \sim N(0, \sigma^2 \mathbb{I}_d)$. Then, $\mathcal{A}$ satisfies $(\alpha, \alpha \|q(X_i)\|_2^2/(2\sigma^2))$-individual RDP for $X_i$.*

**Example 2.7** (Lipschitz analyses). *Suppose that $g : (\mathbb{R}^d)^n \to \mathbb{R}^{d'}$ is $L_i$-Lipschitz in coordinate $i$ (in $\ell_2$-norm). For $q \colon \mathcal{X} \to \mathbb{R}^d$, let $\mathcal{A}(S) = g(q(X_1), \ldots, q(X_n)) + \xi$, $\xi \sim N(0, \sigma^2 \mathbb{I}_{d'})$. Assume that for some $X^\star$, $q(X^\star)$ is the origin. Then, if $S^{-i} = (X_1, \ldots, X_{i-1}, X^\star, X_{i+1}, \ldots, X_n)$, we get that $\mathcal{A}$ satisfies $(\alpha, \alpha L_i^2 \|q(X_i)\|_2^2/(2\sigma^2))$-individual RDP for $X_i$.*

## 3 Fully adaptive composition for Rényi differential privacy

Our main technical contribution is a new adaptive composition theorem for RDP, which bounds the overall privacy loss in terms of the individual privacy losses of all data points. In what follows, we first state a general version of our main theorem, which bounds the privacy loss in adaptive composition in terms of a bound on the sequence of possibly adaptive privacy parameters.

We let $\rho_t$ denote the RDP parameter of order $\alpha$ of $\mathcal{A}_t$, conditional on the past reports. For the sake of generality and simplicity of exposition, we introduce an abstract space $\mathcal{S}$ over pairs of datasets and let

$$\rho_t \stackrel{\text{def}}{=} \sup_{(S,S') \in \mathcal{S}} D_\alpha^\leftrightarrow \left( \mathcal{A}_t(a^{(t-1)}, S) \| \mathcal{A}_t(a^{(t-1)}, S') \right). \tag{1}$$

Typically $\rho_t$ will be random due to the randomness in $a^{(t-1)}$. In the context of individual privacy, we will set $\mathcal{S}$ to be the space of all dataset pairs where either dataset is obtained by deleting $X_i$ from the other. In usual RDP, $\mathcal{S}$ will be the space of all pairs of datasets that differ in any one element.

Theorem 3.1 states that, as long as $\sum_{t=1}^k \rho_t$ is maintained under a fixed budget, the output of adaptive composition preserves privacy.

**Theorem 3.1.** *Fix any $B \geq 0, \alpha \geq 1$, and a set of pairs of datasets $\mathcal{S}$. For any sequence of algorithms $\mathcal{A}_1, \ldots, \mathcal{A}_k$, if $\sum_{t=1}^k \rho_t \leq B$ holds almost surely, where the sequence $\rho_1, \ldots, \rho_k$ is defined in eq. (1), then for all $(S, S') \in \mathcal{S}$ the adaptive composition $\mathcal{A}^{(k)}$ satisfies $D_\alpha^\leftrightarrow \left( \mathcal{A}^{(k)}(S) \| \mathcal{A}^{(k)}(S') \right) \leq B$.*

The proof relies on showing that for any pair $(S, S') \in \mathcal{S}$, $\mathrm{Loss}^{(t)}(a^{(t)}; S, S', \alpha) e^{-(\alpha-1)\sum_{j=1}^t \rho_j}$ — where $\mathrm{Loss}^{(t)}(a^{(t)}; S, S', \alpha)$ denotes the privacy loss incurred up to time $t$, formally defined in the Appendix—is a supermartingale. A related argument is presented in [5] (see Lemma 8), who study privacy composition when a pre-specified set of concentrated DP parameters is adaptively ordered.

We now instantiate Theorem 3.1 in the context of individual privacy. For a fixed point $X$, we let $\mathcal{S}(X, n)$ denote the set of all dataset pairs $(S, S')$ such that $|S| \leq n$ and $S'$ is obtained by deleting element $X$ from $S$. More precisely, $(S, S') \in \mathcal{S}(X, n)$ if $S = (X_1, \ldots, X_m)$, where $m \leq n$ and $X_i = X$ for some $i$, and $S' = S^{-i}$. We use $\rho_t^{(i)}$ to denote the individual privacy parameter of the $t$-th adaptively composed algorithm $\mathcal{A}_t$ with respect to $X_i$, conditional on the past reports:

$$\rho_t^{(i)} \stackrel{\text{def}}{=} \sup_{(S,S') \in \mathcal{S}(X_i, n)} D_\alpha^\leftrightarrow(\mathcal{A}_t(a^{(t-1)}, S) \| \mathcal{A}_t(a^{(t-1)}, S')). \tag{2}$$

Since $\rho_t^{(i)}$ is an instance of definition (1), we can state a direct corollary of Theorem 3.1. Notice that it provides a *data-specific* criterion, while classical composition considers *all* hypothetical datasets.

**Corollary 3.2.** *Fix any $B \geq 0$, $\alpha \geq 1$. If for any input dataset $S = (X_1, \ldots, X_n)$, $\sum_{t=1}^k \rho_t^{(i)} \leq B$ holds almost surely for all individuals $i \in [n]$, then the adaptive composition $\mathcal{A}^{(k)}$ is $(\alpha, B)$-RDP.*

In the following section we show how Corollary 3.2 can be operationalized.

## 4 Rényi privacy filter

Fully adaptive composition was first studied in [29]. They defined the notion of a *privacy filter*, a function that takes as input adaptively-chosen DP parameters $\epsilon_1, \delta_1, \ldots, \epsilon_t, \delta_t$, as well as a global DP budget $\epsilon_g, \delta_g$, and outputs CONT if the overall report after $t$ rounds of adaptive composition with the corresponding privacy parameters is guaranteed to satisfy $(\epsilon_g, \delta_g)$-DP. Otherwise, it outputs HALT.

We show that Theorem 3.1 immediately implies a simple RDP analogue of a privacy filter. Specifically, we show that simply adding up privacy parameters, as in the usual composition where all privacy parameters are fixed up front, is a valid filter for RDP. First we define an RDP filter formally. This general definition is used primarily to explain the relationship of our results to the notions and results in [29]. Our individual privacy filtering application can be derived from Theorem 3.1 directly. As in equation (1), we let $\rho_t$ denote the possibly random RDP parameter of order $\alpha$ of $\mathcal{A}_t$, conditional on the past reports, defined over an implicit space $\mathcal{S}$ of pairs of datasets.

---

**Algorithm 1** Adaptive composition with Rényi privacy filtering

---

**input :** dataset $S \in \mathcal{X}^n$, maximum number of rounds $N \in \mathbb{N}$, sequence of algorithms $(\mathcal{A}_k)_{k=1}^N$

  Initialize $k = 0$

  **while** $k < N$ **do**

      Compute $\rho_{k+1} = \sup_{(S_1,S_2)\in\mathcal{S}} D_\alpha^\leftrightarrow \left(\mathcal{A}_{k+1}(a_1,\ldots,a_k,S_1)\|\mathcal{A}_{k+1}(a_1,\ldots,a_k,S_2)\right)$

      **if** $\mathcal{F}_{\alpha,B}\left(\rho_1,\ldots,\rho_{k+1}\right) = \text{HALT}$ **then**

      |  BREAK

      **end**

      Compute $a_{k+1} = \mathcal{A}_{k+1}(a_1,\ldots,a_k,S)$, set $k \leftarrow k+1$

  **end**

Return $\mathcal{A}^{(k)}(S) = (a_1,\ldots,a_k)$

---

Let $S_\infty$ denote the set of all positive, real-valued finite sequences.

**Definition 4.1** (RDP filter). *Fix $\alpha \geq 1, B \geq 0$. We say that $\mathcal{F}_{\alpha,B} : S_\infty \to \{\text{CONT},\text{HALT}\}$ is a valid Rényi privacy filter, or RDP filter for short, if for any sequence $(\mathcal{A}_k)_{k=1}^N$ and any pair of datasets $(S_1, S_2)$, Algorithm 1 with $\mathcal{S} = \{(S_1, S_2)\}$ satisfies $D_\alpha^\leftrightarrow \left(\mathcal{A}^{(k)}(S_1)\|\mathcal{A}^{(k)}(S_2)\right) \leq B$.*

Without loss of generality, we can assume the filter is monotone, namely that if $\mathcal{F}_{\alpha,B}\left(\rho_1,\ldots,\rho_k\right) = \text{CONT}$ then $\mathcal{F}_{\alpha,B}\left(\rho_1',\ldots,\rho_k'\right) = \text{CONT}$ whenever $\rho_i' \leq \rho_i, \forall i \in [k]$. This implies that an RDP filter can be applied with $\rho_k$ defined using an arbitrary set $\mathcal{S}$ instead of just a single pair of datasets.

**Lemma 4.2.** *Fix $\alpha \geq 1, B \geq 0$. If $\mathcal{F}_{\alpha,B} : S_\infty \to \{\text{CONT},\text{HALT}\}$ is a valid RDP filter, then for any sequence $(\mathcal{A}_k)_{k=1}^N$ and any set $\mathcal{S}$, Algorithm 1 has $\sup_{(S_1,S_2)\in\mathcal{S}} D_\alpha^\leftrightarrow \left(\mathcal{A}^{(k)}(S_1)\|\mathcal{A}^{(k)}(S_2)\right) \leq B$.*

We remark that the analyst might choose an algorithm at time $t$ that exceeds the privacy budget, which will trigger the filter $\mathcal{F}_{\alpha,B}$ to halt. However, the analyst can then decide to change the computation at time $t$ retroactively and query the filter again, which then might allow continuation. This way, one can ensure a sequence of $N$ computations with formal privacy guarantees, for any target number of rounds $N$. In the following subsection, we present an application of RDP filters to individual privacy loss accounting which relies on this reasoning.

**Theorem 4.3.** *Let*

$$\mathcal{F}_{\alpha,B}(\rho_1,\ldots,\rho_k) = \begin{cases} \text{CONT}, & \text{if } \sum_{t=1}^k \rho_t \leq B, \\ \text{HALT}, & \text{if } \sum_{t=1}^k \rho_t > B. \end{cases}$$

*Then, $\mathcal{F}_{\alpha,B}$ is a valid Rényi privacy filter.*

**Remark 4.4.** *For algorithms that satisfy zero-concentrated DP (zCDP), the stopping rule of Theorem 4.3 suffices for controlling zCDP privacy loss as well. Namely, if $\mathcal{A}_t(a_1,\ldots,a_{t-1},\cdot)$ is $\kappa_t$-zCDP, then the halting criterion of the Rényi privacy filter with parameters $(\alpha, \alpha\kappa)$ is $\sum_{t=1}^k \alpha\kappa_t \leq \alpha\kappa$, which simplifies to $\sum_{t=1}^k \kappa_t \leq \kappa$. Since this stopping rule is independent of $\alpha$, we conclude that the overall output is $(\alpha, \alpha\kappa)$-RDP for all $\alpha \geq 1$, which is equivalent to $\kappa$-zCDP. More generally, Theorem 4.3 extends to tracking Rényi privacy loss for any set of orders $\{\alpha_j\}_{j\in\mathcal{I}}$: if all filters in the set $\{\mathcal{F}_{\alpha_j,B}\}_{j\in\mathcal{I}}$ output $\text{CONT}$, then the adaptive composition satisfies $(\alpha_j, B)$-RDP, $\forall j \in \mathcal{I}$.*

## 4.1 Individual privacy accounting via a privacy filter

Now we design an *individual privacy filter*, which monitors individual privacy loss estimates across all individuals and all computations, and ensures that the privacy of all individuals is preserved. The filter guarantees privacy by adaptively dropping data points once their cumulative individual privacy

loss estimate is about to cross a pre-specified budget. More specifically, at every step of adaptive composition $t$, it determines an active set of points $S_t \subseteq S$ based on cumulative estimated individual losses, and applies $\mathcal{A}_t$ only to $S_t$.

---

**Algorithm 2** Adaptive composition with individual privacy filtering

---

**input :** dataset $S \in \mathcal{X}^n$, sequence of algorithms $(\mathcal{A}_t)_{t=1}^k$
**for** $t = 1, \ldots, k$ **do**

> For all $X_i \in S$, compute $\rho_t^{(i)} = \sup_{(S_1, S_2) \in \mathcal{S}(X_i, n)} D_\alpha^{\leftrightarrow}\left(\mathcal{A}_t(a^{(t-1)}, S_1) \| \mathcal{A}_t(a^{(t-1)}, S_2)\right)$
> Determine active set $S_t = (X_i \; : \; \mathcal{F}_{\alpha, B}(\rho_1^{(i)}, \ldots, \rho_t^{(i)}) = \text{CONT})$
> For all $X_i \in S$, set $\rho_t^{(i)} \leftarrow \rho_t^{(i)} \mathbf{1}\{X_i \in S_t\}$
> Compute $a_t = \mathcal{A}_t(a_1, \ldots, a_{t-1}, S_t)$

**end**
Return $(a_1, \ldots, a_k)$

---

Here, $\mathcal{F}_{\alpha, B}$ is the filter from Theorem 4.3. Its validity implies that Algorithm 2 preserves RDP.

**Theorem 4.5.** *Adaptive composition with individual privacy filtering (Alg. 2) satisfies $(\alpha, B)$-RDP.*

**Remark 4.6.** *Algorithm 2 can naturally be applied to privacy accounting in the local DP model. Here, each user would have a local implementation of the individual privacy filter and would stop responding when the filter halts. This is possible because the decision to halt or continue for any data point does not depend on the other data points other than through the sequence of reports $a^{(t)}$.*

## 4.2   Answering linear queries

To illustrate the gains of individual privacy, we consider the task of answering adaptively-chosen high-dimensional linear queries. We aim to design an algorithm that receives a sequence of queries $q_1, q_2, \ldots$, where $q_t : \mathcal{X} \to \mathbb{R}^d$ for all $t \in \mathbb{N}$, and upon receiving $q_t$ provides an estimate $a_t$ of $\sum_{i=1}^n q_t(X_i)$, where $S = (X_1, \ldots, X_n) \in \mathcal{X}^n$. In some applications it is natural to expect that, for a typical user, many of the queries evaluate to a very small value (having norm close to zero). For example, in the context of continual monitoring [15, 20], a platform might collect one real-valued indicator per user per day, and wish to make decisions based off the daily averages of these indicators across users. Here, $X_i$ would be a single user, and $q_t(X_i) = X_t^{(i)}$ would be the corresponding user's indicator on day $t$. For example, $X_t^{(i)} \in \{0, 1\}$ could be a binary indicator of a change of some state for user $i$ on day $t$. For simplicity, we will treat the dataset $S$ as fixed, but our results apply to a more general setting in which the users' data can be updated after each query; for example, additional points might arrive in the process of the analysis (see, e.g., [7]).

A prototypical mechanism for answering linear queries is the Gaussian mechanism, which reports $a_t = \sum_{i=1}^n q_t(X_i) + \xi_t$, where $\xi_t \sim N(0, \sigma^2 \mathbb{I}_d)$. If the range of $q_t$ is constrained (or clipped) to have norm at most $C$, then the worst-case RDP loss incurred by answering $q_t$ is $(\alpha, \rho_t) = (\alpha, \alpha C^2/(2\sigma^2))$. This implies that the standard analysis—which only considers $\rho_t$—allows answering at most $k_0 = \lfloor 2B\sigma^2/(C^2\alpha) \rfloor$ queries in order to ensure $(\alpha, B)$-RDP. As we mentioned in Example 2.6, the Gaussian mechanism satisfies $(\alpha, \rho_t^{(i)})$-individual RDP for $X_i$, where

$$\rho_t^{(i)} = \alpha \|q_t(X_i)\|_2^2/(2\sigma^2).$$

Thus the individual privacy filter allows us to provide accurate answers to $q_t$ as long as each user's responses are "sparse" (more generally, have small $\sum_{j=1}^t \|q_j(X_i)\|_2^2$). Formally, we obtain the following generalization of Theorem 1.2.

**Corollary 4.7.** *There exists an algorithm $\mathcal{A}$ that, given a norm budget $B_{\text{norm}}$ and privacy level $\kappa$, for any adaptively-chosen sequence of queries $q_1, \ldots, q_k$ of arbitrary length $k$, where $q_i : \mathcal{X} \to \mathbb{R}^d$, provides a sequence of answers $a_1, \ldots, a_k$ such that: (1) $\mathcal{A}$ is $\kappa$-zCDP, that is, $(\alpha, \alpha\kappa)$-RDP for all $\alpha \geq 1$; (2) for all $t$ and any $\delta \in (0, 1)$, the probability that $\|a_t - \sum_{X_i \in S_t} q_t(X_i)\|_\infty > \sqrt{B_{\text{norm}} \log(d/\delta)}/\kappa$ is at most $\delta$, where $S_t = (X_i \in S : \sum_{j=1}^t \|q_j(X_i)\|_2^2 \leq B_{\text{norm}})$.*

Corollary 4.7 follows from Theorem 4.5, by setting each $\mathcal{A}_t$ to be the Gaussian mechanism with $\sigma^2 = B_{\text{norm}}/(2\kappa)$. Note that, due to $\rho_t^{(i)} \leq \rho_t$, *all* points are active in $S_t$ for at least the first $k_0$

computations, as prescribed by the usual worst-case analysis, and during those $k_0$ steps the answers are guaranteed to be accurate. Therefore, individual privacy provides a more fine-grained way of quantifying privacy loss by taking into account the value of the point whose loss we aim to measure. While $k$ is technically allowed to be arbitrarily large, after a certain number of reports we expect few points to remain active; we discuss stopping criteria in the following section.

### 4.3 Filter for $(\epsilon, \delta)$-differential privacy

By connections between RDP and DP [4, 26], we can translate our Rényi privacy filter into a filter for approximate DP. We define a valid DP filter analogously to a valid RDP filter, the difference being that it takes as input DP, rather than RDP parameters, and that it is parameterized by a global DP budget $\epsilon_g \geq 0, \delta_g \in (0, 1)$. We denote by $\epsilon_t$ the possibly adaptive DP parameter of $\mathcal{A}_t$,

$$\epsilon_t \overset{\text{def}}{=} \sup_{(S_1, S_2) \in \mathcal{S}} D_\infty^\leftrightarrow \left( \mathcal{A}_t(a^{(t-1)}, S_1) \| \mathcal{A}_t(a^{(t-1)}, S_2) \right).$$

Here, $D_\infty$ denotes the max-divergence, obtained as the limit of Rényi divergence by taking $\alpha \to \infty$. We also denote a DP filter by $\mathcal{G}_{\epsilon_g, \delta_g}$.

We focus on advanced composition of *pure* DP algorithms $\mathcal{A}_t$. As shown in [29] (see Lemma 3.3 in the latest arXiv version), a DP filter for approximately DP algorithms $\mathcal{A}_t$ can be obtained by an extension of a filter for pure DP.

Our analysis implies a simple stopping condition for a DP filter, in terms of any zCDP level which ensures $(\epsilon_g, \delta_g)$-DP. For clarity, we give one particularly simple translation from zCDP to DP, however one could in principle invoke more sophisticated analyses such as those of Bun and Steinke [4].

**Theorem 4.8.** *Let $B^\star$ be any $B \geq 0$ such that $B$-zCDP implies $(\epsilon_g, \delta_g)$-DP. Then,*

$$\mathcal{G}_{\epsilon_g, \delta_g}(\epsilon_1, \ldots, \epsilon_k) = \begin{cases} \text{CONT}, & \text{if } \frac{1}{2} \sum_{t=1}^k \epsilon_t^2 \leq B^\star, \\ \text{HALT}, & \text{if } \frac{1}{2} \sum_{t=1}^k \epsilon_t^2 > B^\star \end{cases}$$

*is a valid DP filter. For example, we can take $B^\star = (-\sqrt{\log(1/\delta_g)} + \sqrt{\log(1/\delta_g) + \epsilon_g})^2$.*

For example, if the privacy parameters are fixed up front and $\epsilon_t \equiv \epsilon$, simplifying the stopping criterion of the above filter implies that the adaptive composition of $k$ $\epsilon$-DP algorithm satisfies $(\frac{1}{2}k\epsilon^2 + \sqrt{2k \log(1/\delta)}\epsilon, \delta)$-DP, for all $\delta > 0$. This tightens the rate of Rogers et al. [29], whose filter halts when

$$\frac{1}{2}k\epsilon(e^\epsilon - 1) + \sqrt{2\left(k\epsilon^2 + C(\epsilon_g, \delta_g)\right)\left(1 + 0.5 \log\left(1 + \frac{k\epsilon^2}{C(\epsilon_g, \delta_g)}\right)\right) \log(1/\delta_g)} > \epsilon_g,$$

where $C(\epsilon_g, \delta_g) = \frac{\epsilon_g^2}{28.04 \log(1/\delta_g)}$. The factor $C(\epsilon_g, \delta_g)$ essentially determines the gap between our analysis and the analysis of Rogers et al., and our filter is noticeably tighter for non-negligible values of $C(\epsilon_g, \delta_g)$. In addition, our filter has an arguably simpler stopping criterion.

Further improvements on the rate are possible via a more intricate conversion between zCDP and DP, as presented in [4].

## 5 Private gradient descent with individual privacy accounting

A popular approach to private model training via gradient descent is to clip the norm of all gradients at every step and add Gaussian noise to the clipped gradients [1]. Existing analyses compute the privacy spent so far by using a uniform upper bound on the gradient norms, determined by the clipping value. Using the individual privacy filter, we develop a less conservative version of this approach, one which takes into account the *realized* norms of the gradients, rather than just their upper bound.

There are various ways to incorporate individual privacy filtering into private gradient descent (GD) [1]. To facilitate the comparison, we present a particularly simple one. As in private GD, at every step we clip all computed gradients and add Gaussian noise. However, after the round at which private GD would halt, we look at the "leftover" privacy budget for all points, and utilize them until their budget

runs out. The leftover budget is essentially the gap between the worst-case sum of squared gradient norms (determined by the clipping value) and the sum of squared norms of the *realized* gradients.

---

**Algorithm 3** Private gradient descent with filtering

---

**input :** dataset $(X_1, \ldots, X_n)$, loss function $\ell(\theta; X_i)$, learning rate $(\eta_t)_{t=1}^{\infty}$, noise scale $\sigma > 0$, clip value $C > 0$, number of steps $k_{\max} \in \mathbb{N}$, squared norm budget $B_{\mathrm{norm}} > 0$

Initialize $\theta_1$ arbitrarily

  **for** $t = 1, 2, \ldots, k_{\max}$ **do**

      Compute gradients $g_t(X_i) \leftarrow \nabla_\theta \ell(\theta_t; X_i), \forall i$

      Clip $\bar{g}_t(X_i) \leftarrow \frac{g_t(X_i)}{\|g_t(X_i)\|_2} \cdot \min \left\{ \|g_t(X_i)\|_2, \; C, \; \sqrt{B_{\mathrm{norm}} - \sum_{j=1}^{t-1} \|\bar{g}_j(X_i)\|_2^2} \right\}, \forall i$

      Add noise $\widetilde{g}_t \leftarrow \frac{1}{n} \sum_{i=1}^{n} (\bar{g}_t(X_i) + N(0, \sigma^2 C^2 \mathbb{I}))$

      Take gradient step $\theta_{t+1} \leftarrow \theta_t - \eta_t \widetilde{g}_t$

**end**

Return $\theta_{k+1}$

---

For at least $\lfloor B_{\mathrm{norm}}/C^2 \rfloor$ rounds, all gradients get clipped to have norm at most $C$. After that, points adaptively get filtered out once the accumulated squared norm of their (clipped) gradients hits $B_{\mathrm{norm}}$.

**Proposition 5.1.** *Algorithm 3 satisfies $\frac{B_{\mathrm{norm}}}{2\sigma^2 C^2}$-zCDP, or, equivalently, $\left(\alpha, \frac{\alpha B_{\mathrm{norm}}}{2\sigma^2 C^2}\right)$-RDP for all $\alpha \geq 1$.*

When $B_{\mathrm{norm}} = kC^2$, the guarantees of Algorithm 3 are the same as those of private GD with $k$ steps. However, they *do not* depend on the total number of steps $k_{\max}$ ($k_{\max}$ need not be equal to $k$), which raises the question of how to set $k_{\max}$. (Certainly $k_{\max}$ should be at least $\lfloor B_{\mathrm{norm}}/C^2 \rfloor$, otherwise the privacy budget is not used up fully for any data point.) If $k_{\max}$ is small, we might stop the optimization too early and thus forgo a potentially higher accuracy; if $k_{\max}$ is large, then a lot of points might get filtered out and we might add high amounts of noise relative to the number of active points. One solution is to periodically estimate the number of active points in a privacy-preserving fashion. After round $\lfloor B_{\mathrm{norm}}/C^2 \rfloor$, the analyst can estimate the size of the active set $\{i : \sum_{j=1}^{t} \|\bar{g}_j(X_i)\|_2^2 \leq B_{\mathrm{norm}}\}$ (which is a simple linear query) and use it to stop. To reduce the privacy cost of such estimates one can use the continual monitoring technique [15] since each point is filtered out only once. Alternatively, if one only wants to ensure that the size of the active set exceeds a fixed threshold, one can use the sparse vector technique [14, 10] and thus incur an even smaller privacy loss due to adaptive stopping. Another solution is to periodically check the training accuracy, which is again a linear query, and stop once it plateaus.

### 5.1 Experiments

As proof of concept, we compare the performance of private GD and its generalization with filtering (Algorithm 3) by training a convolutional neural network on MNIST [24]. We use the default architecture in the MNIST example of the PyTorch Opacus library [33].

We remark that, in practice, it is more common to use private SGD, rather than batch GD. While in principle it is possible to compute individual privacy parameters for SGD, random subsampling of points requires computing gradients for *all* points at every step to observe gains from individual accounting. As a result, SGD is no less computationally expensive than batch GD in the context of individual accounting. Nevertheless, GD requires fewer steps and—importantly— we observe that it achieves a significantly better privacy-utility tradeoff. For example, using the same architecture, the Opacus library reports accuracy $(94.63\% \pm 0.34\%)$ for $\epsilon = 1.16$ and $\delta = 10^{-5}$, which is almost the same accuracy we obtain with $\epsilon = 0.5$ and $\delta = 10^{-5}$ (see table below). Recent large-scale experiments on differentially private training of language models [2] similarly point to larger batches leading to higher utility, and we believe this phenomenon likely holds in many other settings and is worth further exploration.

All reported average accuracies and deviations are estimated over 10 trials. We fix the target DP parameters $(\epsilon, \delta)$, and evaluate the test accuracy. We set $\delta = 10^{-5}$, and vary the value of $\epsilon$. In the first set of evaluations, for every $\epsilon$ we tune all algorithm hyperparameters to achieve high test accuracy with private GD. For private GD with filtering, to make the comparison as clear as possible, we adopt the same hyperparameters. After $B_{\mathrm{norm}}/C^2$ steps, we query the training accuracy a fixed number

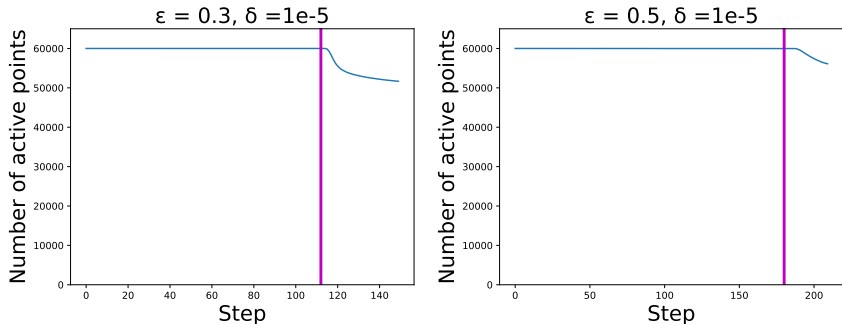

Figure 1: Number of active points during one run of private GD with filtering in the tuned regime, for $\epsilon = 0.3$ (left) and $\epsilon = 0.5$ (right). The solid vertical line denotes step $B_{\mathrm{norm}}/C^2$.

of times in intervals of 5 steps and stop adaptively, when the training accuracy is observed to be the highest. We provide the specifics of the stopping rule and other experimental details in the Appendix.

| $\epsilon$ | GD (tuned) | GD (tuned) w/ filtering | GD (suboptimal) | GD (suboptimal) w/ filtering |
|---|---|---|---|---|
| 0.3 | $(93.29 \pm 0.49)\%$ | $(93.64 \pm 0.46)\%$ | $(84.47 \pm 3.95)\%$ | $(92.25 \pm 0.91)\%$ |
| 0.5 | $(94.62 \pm 0.43)\%$ | $(94.90 \pm 0.26)\%$ | $(92.07 \pm 2.07)\%$ | $(94.30 \pm 0.58)\%$ |
| 1.0 | $(96.25 \pm 0.23)\%$ | $(96.25 \pm 0.23)\%$ | $(94.45 \pm 0.45)\%$ | $(95.33 \pm 0.34)\%$ |

Overall we observe modest accuracy improvements with individual filtering in the tuned regime. The benefits are more noticeable for small $\epsilon$, while for large $\epsilon$, the accuracy plateaus after $B_{\mathrm{norm}}/C^2$ steps and hence we do not add extra steps. In Figure 1 we plot the number of active points, i.e. those that have not yet exhausted their privacy budget, for $\epsilon \in \{0.3, 0.5\}$. Due to extensive hyperparameter tuning in this specific application, private GD is implicitly tuned to clip the gradients in such a way that hard-to-classify points exhaust their privacy budget. Such tuning, however, is not possible when queries are chosen by human analysts (which is also hard to run experiments on) and in federated settings where data is held by the clients and finding the optimal setting of hyperparameters is typically infeasible. Therefore we examine the benefits of individual filtering in examples of such suboptimally tuned settings. For example, we evaluate the performance when the clipping value $C$ is chosen to be larger than in the optimal setting (while keeping the noise level the same). Specifically, for $\epsilon \in \{0.3, 0.5\}$, we set $C$ to be $1.5$ the optimally tuned value, and for $\epsilon = 1.0$ we set $C$ to be double the tuned value. We reduce the number of optimization steps accordingly to achieve the same privacy guarantee. In the suboptimal regime, the benefits of individual filtering become much more significant as a large fraction of points remain in the active pool after $B_{\mathrm{norm}}/C^2$ steps. Similar results are obtained when the noise rate is not set optimally and we include the results in the Appendix.

Altogether, we view this application as a useful proof of concept: it demonstrates that individual accounting is practical, easy to implement, and can only make the results better.

## Acknowledgements

We thank Katrina Ligett, Kunal Talwar, and Neil Vexler for insightful discussions on individual notions of privacy and feedback on this work. We are grateful to Ryan McKenna for providing code and suggestions for the experiments. TZ acknowledges support from an Apple PhD Fellowship.

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
