# A  Proofs

## A.1  Proof of Theorem 3.1

First we set up some notation. For two datasets $S$ and $S'$, parameter $\alpha \geq 1$ and *fixed* $a^{(t)}$, we let[1]

$$\text{Loss}^{(t)}(a^{(t)}; S, S', \alpha) \stackrel{\text{def}}{=} \left( \frac{\Pr\left[\mathcal{A}^{(t)}(S) = a^{(t)}\right]}{\Pr\left[\mathcal{A}^{(t)}(S') = a^{(t)}\right]} \right)^{\alpha}.$$

Similarly, for fixed $a^{(t)}$ we also define

$$\text{Loss}_t(a^{(t)}; S, S', \alpha) \stackrel{\text{def}}{=} \left( \frac{\Pr\left[\mathcal{A}_t(a_1, \ldots, a_{t-1}, S) = a_t\right]}{\Pr\left[\mathcal{A}_t(a_1, \ldots, a_{t-1}, S') = a_t\right]} \right)^{\alpha}.$$

Roughly speaking, $\text{Loss}^{(t)}$ denotes the total privacy loss incurred by the first $t$ rounds of adaptive composition, while $\text{Loss}_t$ denotes the loss incurred in round $t$ (which, due to adaptivity, depends on the outcomes of the first $t-1$ rounds).

Note that, since

$$\Pr\left[\mathcal{A}^{(t)}(S) = a^{(t)}\right] = \Pr\left[\mathcal{A}^{(t-1)}(S) = a^{(t-1)}\right] \Pr\left[\mathcal{A}_t(a_1, a_2, \ldots, a_{t-1}, S) = a_t\right],$$

we have $\text{Loss}^{(t)}(a^{(t)}; S, S', \alpha) = \text{Loss}^{(t-1)}(a^{(t-1)}; S, S', \alpha) \cdot \text{Loss}_t(a^{(t)}; S, S', \alpha)$.

Fix any $(S, S') \in \mathcal{S}$. In what follows, we take $a^{(t)} = (a_1, \ldots, a_t)$ to be distributed as the random output of adaptive composition applied to $S'$, that is $a^{(t)} \sim \mathcal{A}^{(t)}(S')$. Consequently, $\text{Loss}^{(t)}(a^{(t)}; S, S', \alpha)$ and $\text{Loss}_t(a^{(t)}; S, S', \alpha)$ are also random.

Let $M_t \stackrel{\text{def}}{=} \text{Loss}^{(t)}(a^{(t)}; S, S', \alpha)e^{-(\alpha-1)\sum_{j=1}^t \rho_j}$, and let $M_0 = 1$. Consider the filtration $\Gamma_t = \sigma(a^{(t)})$. We prove that $M_t$ is a supermartingale with respect to $\Gamma_t$; that is, we show $\mathbb{E}[M_t \mid \Gamma_{t-1}] \leq M_{t-1}$. This follows since:

$$\mathbb{E}[M_t \mid \Gamma_{t-1}] = \mathbb{E}\left[ \text{Loss}^{(t)}(a^{(t)}; S, S', \alpha)\, e^{-(\alpha-1)\sum_{j=1}^t \rho_j} \,\middle|\, \Gamma_{t-1} \right]$$

$$= \mathbb{E}\left[ \text{Loss}^{(t-1)}(a^{(t-1)}; S, S', \alpha)\, \text{Loss}_t(a^{(t)}; S, S', \alpha)\, e^{-(\alpha-1)\sum_{j=1}^t \rho_j} \,\middle|\, \Gamma_{t-1} \right]$$

$$= \text{Loss}^{(t-1)}(a^{(t-1)}; S, S', \alpha)\, e^{-(\alpha-1)\sum_{j=1}^t \rho_j} \mathbb{E}\left[ \text{Loss}_t(a^{(t)}; S, S', \alpha) \,\middle|\, \Gamma_{t-1} \right]$$

$$\leq \text{Loss}^{(t-1)}(a^{(t-1)}; S, S', \alpha)\, e^{-(\alpha-1)\sum_{j=1}^t \rho_j} e^{(\alpha-1)\rho_t}$$

$$= \text{Loss}^{(t-1)}(a^{(t-1)}; S, S', \alpha)\, e^{-(\alpha-1)\sum_{j=1}^{t-1} \rho_j}$$

$$= M_{t-1},$$

where the third equality uses the fact that $(\rho_j)_{j=1}^t \in \Gamma_{t-1}$, and the inequality applies the definition of $\rho_t$. Therefore, by applying iterated expectations, we can conclude

$$\mathbb{E}[M_k] = \mathbb{E}_{a^{(k)} \sim \mathcal{A}^{(k)}(S')} \left[ \text{Loss}^{(k)}\left(a^{(k)}; S, S', \alpha\right) e^{-(\alpha-1)\sum_{j=1}^k \rho_j} \right] \leq \mathbb{E}[M_0] = 1.$$

Since $\sum_{j=1}^k \rho_j \leq B$ by assumption, this inequality implies

$$\mathbb{E}_{a^{(k)} \sim \mathcal{A}^{(k)}(S')} \left[ \text{Loss}^{(k)}\left(a^{(k)}; S, S', \alpha\right) \right] \leq e^{(\alpha-1)B}.$$

After normalizing, we get

$$D_\alpha\left(\mathcal{A}^{(k)}(S), \mathcal{A}^{(k)}(S')\right) = \frac{1}{\alpha - 1} \log \mathbb{E}_{a^{(k)} \sim \mathcal{A}^{(k)}(S')} \left[ \text{Loss}^{(k)}\left(a^{(k)}; S, S', \alpha\right) \right] \leq B.$$

The same argument can be used to bound the other direction of the divergence. Since the choice of $(S, S')$ was arbitrary, we can conclude $\sup_{(S,S')\in\mathcal{S}} D_\alpha^{\leftrightarrow}\left(\mathcal{A}^{(k)}(S), \mathcal{A}^{(k)}(S')\right) \leq B$, as desired.

---

[1] All algorithms we are considering, if not discrete, induce a density w.r.t. the Lebesgue measure. For such instances, replacing expressions such as $\Pr\left[\mathcal{A}^{(t)}(S) = a\right]$ with the density of $\mathcal{A}^{(t)}(S)$ at $a$ gives the analysis in the continuous case.

## A.2 Proof of Corollary 3.2

By Theorem 3.1, $\sum_{t=1}^{k} \rho_t^{(i)} \le B$ implies that

$$D_\alpha^\leftrightarrow \left( \mathcal{A}^{(k)}(S) \| \mathcal{A}^{(k)}(S^{-i}) \right) \le B.$$

Since this holds for all $S \in \mathcal{X}^n$ and $i \in [n]$, we conclude that $\mathcal{A}^{(k)}$ is $(\alpha, B)$-Rényi differentially private.

## A.3 Proof of Theorem 4.3

The only difference between Theorem 3.1 and this theorem is that a privacy filter halts at a random round, meaning the length of the output is random rather than fixed. Therefore, in this proof we formalize the fact that Theorem 3.1 is valid even under adaptive stopping.

Fix any $(S_1, S_2)$. By the argument in Theorem 3.1, $M_t \stackrel{\text{def}}{=} \text{Loss}^{(t)}(a^{(t)}; S_1, S_2, \alpha)e^{-(\alpha-1)\sum_{j=1}^{t} \rho_j}$, where $a^{(t)} \sim \mathcal{A}^{(t)}(S_2)$, is a supermartingale with respect to $\Gamma_t = \sigma(a^{(t)})$.

Let $T$ be the last round before Algorithm 1 halts; that is,

$$T = \min \left\{ t : \mathcal{F}_{\alpha,B}(\rho_1, \ldots, \rho_{t+1}) = \text{HALT} \right\} \wedge N.$$

Note that $T$ is a stopping time with respect to $\Gamma_t$, that is $\{T = t\} \in \Gamma_t$, due to the fact that $\rho_{t+1} \in \Gamma_t$. Since $T$ is almost surely bounded by construction, we can apply the optional stopping theorem for supermartingales to get

$$\mathbb{E}[M_T] = \mathbb{E}_{a^{(T)} \sim \mathcal{A}^{(T)}(S_2)} \left[ \text{Loss}^{(T)}\left(a^{(T)}; S_1, S_2, \alpha\right) e^{-(\alpha-1)\sum_{j=1}^{T} \rho_j} \right] \le \mathbb{E}[M_0] = 1.$$

By definition of the RDP filter, we know that $\sum_{j=1}^{T} \rho_j \le B$ almost surely; otherwise the filter would have halted earlier. Thus, we can conclude

$$\mathbb{E}_{a^{(T)} \sim \mathcal{A}^{(T)}(S_2)} \left[ \text{Loss}^{(T)}\left(a^{(T)}; S_1, S_2, \alpha\right) e^{-(\alpha-1)B} \right] \le 1.$$

After rearranging and normalizing, this implies

$$D_\alpha \left( \mathcal{A}^{(T)}(S_1), \mathcal{A}^{(T)}(S_2) \right) = \frac{1}{\alpha - 1} \log \mathbb{E}_{a^{(T)} \sim \mathcal{A}^{(T)}(S_2)} \left[ \text{Loss}^{(T)}\left(a^{(T)}; S_1, S_2, \alpha\right) \right] \le B.$$

The same argument can be used to bound the other direction of the divergence. Therefore, $\mathcal{F}_{\alpha,B}$ is a valid RDP filter.

## A.4 Proof of Theorem 4.5

Denote by $\mathcal{A}_t^{\text{filt}}$ the subroutine given by the $t$-th step of the individual filtering algorithm; that is, $a_t = \mathcal{A}_t^{\text{filt}}(a_1, \ldots, a_{t-1}, S)$. Note that $\mathcal{A}_t^{\text{filt}}$ is *not* equal to $\mathcal{A}_t$. By analogy with the notation $\mathcal{A}^{(t)}$, we also let $\mathcal{A}^{\text{filt}(t)}(\cdot) \stackrel{\text{def}}{=} (\mathcal{A}_1^{\text{filt}}(\cdot), \mathcal{A}_2^{\text{filt}}(\mathcal{A}_1^{\text{filt}}(\cdot), \cdot), \ldots, \mathcal{A}_t^{\text{filt}}(\mathcal{A}_1^{\text{filt}}(\cdot), \ldots, \cdot))$.

We argue that the privacy loss of point $X_i$ in round $t$, conditional on the past reports, is upper bounded by $\rho_t^{(i)}$ (after $\rho_t^{(i)}$ has been updated):

$$\max \left\{ \mathbb{E}_{a^{(t)} \sim \mathcal{A}^{\text{filt}(t)}(S^{-i})} \left[ \text{Loss}_t^{\text{filt}}(a^{(t)}; S, S^{-i}, \alpha) \mid a^{(t-1)} \right], \mathbb{E}_{a^{(t)} \sim \mathcal{A}^{\text{filt}(t)}(S)} \left[ \text{Loss}_t^{\text{filt}}(a^{(t)}; S^{-i}, S, \alpha) \mid a^{(t-1)} \right] \right\}$$

$$\le e^{(\alpha-1)\rho_t^{(i)}},$$

where $\text{Loss}_t^{\text{filt}}(a^{(t)}; S, S', \alpha) = \left( \frac{\Pr\left[\mathcal{A}_t^{\text{filt}}(a_1, \ldots, a_{t-1}, S) = a_t\right]}{\Pr\left[\mathcal{A}_t^{\text{filt}}(a_1, \ldots, a_{t-1}, S') = a_t\right]} \right)^\alpha$. Similarly, we adopt the definition of $\text{Loss}_t$ from the proof of Theorem 3.1.

To do so, we reason about the active set of points at time $t$ when the input to adaptive composition is $S$, and when the input is $S^{-i}$. Denote by $S_t$ the active set given input $S$, and by $S_t^{(i)}$ the active set given input $S^{-i}$. Observe that, conditional on $a_1, \ldots, a_{t-1}$, we have $(S_t, S_t^{(i)}) \in \mathcal{S}(X_i, n)$; that

is, $S_t$ and $S_t^{(i)}$ differ only in the presence on $X_i$. This follows because the sequence $(\rho_j^{(i)})_{j=1}^t$ is measurable with respect to $a_1, \ldots, a_{t-1}$, and whether point $X_i$ is active at time $t$ is in turn determined based only on $(\rho_j^{(i)})_{j=1}^t$. In particular, whether any given point is active does not depend on the rest of the input dataset ($S$ or $S^{-i}$), given $a_1, \ldots, a_{t-1}$. Therefore, if $X_i \notin S_t$, then $X_i$ loses no privacy in round $t$, because $\mathcal{A}_t^{\text{filt}}(a_1, \ldots, a_{t-1}, S) \stackrel{d}{=} \mathcal{A}_t^{\text{filt}}(a_1, \ldots, a_{t-1}, S^{-i})$, conditional on $a_1, \ldots, a_{t-1}$. On the other hand, if $X_i \in S_t$, then its privacy loss can be bounded as

$$\max \left\{ \mathop{\mathbb{E}}_{a^{(t)} \sim \mathcal{A}^{\text{filt}(t)}(S^{-i})} \left[ \text{Loss}_t^{\text{filt}}(a^{(t)}; S, S^{-i}, \alpha) \,\Big|\, a^{(t-1)} \right], \mathop{\mathbb{E}}_{a^{(t)} \sim \mathcal{A}^{\text{filt}(t)}(S)} \left[ \text{Loss}_t^{\text{filt}}(a^{(t)}; S^{-i}, S, \alpha) \,\Big|\, a^{(t-1)} \right] \right\}$$

$$\leq \max \left\{ \mathop{\mathbb{E}}_{a^{(t)} \sim \mathcal{A}^{\text{filt}(t)}(S^{-i})} \left[ \text{Loss}_t(a^{(t)}; S_t, S_t^{(i)}, \alpha) \,\Big|\, a^{(t-1)} \right], \right.$$
$$\left. \mathop{\mathbb{E}}_{a^{(t)} \sim \mathcal{A}^{\text{filt}(t)}(S)} \left[ \text{Loss}_t(a^{(t)}; S_t^{(i)}, S_t, \alpha) \,\Big|\, a^{(t-1)} \right] \right\}$$

$$\leq \sup_{(S_1, S_2) \in \mathcal{S}(X_i, n)} \max \left\{ \mathop{\mathbb{E}}_{a^{(t)} \sim \mathcal{A}^{(t)}(S_2)} \left[ \text{Loss}_t(a^{(t)}; S_1, S_2, \alpha) \,\Big|\, a^{(t-1)} \right], \right.$$
$$\left. \mathop{\mathbb{E}}_{a^{(t)} \sim \mathcal{A}^{(t)}(S_1)} \left[ \text{Loss}_t(a^{(t)}; S_2, S_1, \alpha) \,\Big|\, a^{(t-1)} \right] \right\}$$

$$\leq e^{(\alpha-1)\rho_t^{(i)}}.$$

With this, we have showed that $\rho_t^{(i)}$ is a valid estimate of the privacy loss of $X_i$, for all $i \in [n]$.

Now we argue that, at the end of every round $t$ (after $\rho_t^{(i)}$ has been updated), $\mathcal{F}_{\alpha, B}(\rho_1^{(i)}, \ldots, \rho_t^{(i)}) = \text{CONT}$ for all $i \in [n]$. This follows by induction. For $t = 1$, this is clearly true because $\mathcal{F}_{\alpha, B}(0) = \text{CONT}$. Now assume it is true at time $t - 1$. Then, at time $t$, the filter clearly continues for all $X_i \in S_t$ simply by definition of $S_t$. If $X_i \notin S_t$, then $\mathcal{F}_{\alpha, B}(\rho_1^{(i)}, \ldots, \rho_t^{(i)}) = \mathcal{F}_{\alpha, B}(\rho_1^{(i)}, \ldots, \rho_{t-1}^{(i)}, 0) = \mathcal{F}_{\alpha, B}(\rho_1^{(i)}, \ldots, \rho_{t-1}^{(i)}) = \text{CONT}$. Therefore, we conclude that at the end of every round $t \in [k]$ and for all $i \in [n]$, the filter would output CONT. By the validity of $\mathcal{F}_{\alpha, B}$, we know that $\mathcal{F}_{\alpha, B}(\rho_1^{(i)}, \ldots, \rho_t^{(i)}) = \text{CONT}$ implies

$$D_\alpha^\leftrightarrow \left( \mathcal{A}^{(t)}(S) \| \mathcal{A}^{(t)}(S^{-i}) \right) \leq B,$$

and since this holds for all $S$ and all $i \in [n]$, we conclude that Algorithm 2 is $(\alpha, B)$-RDP.

## A.5   Proof of Theorem 4.8

By conversions between DP and zCDP [4], we know that $\epsilon_t$-DP implies $\frac{1}{2}\epsilon_t^2$-zCDP, that is $(\alpha, \frac{1}{2}\epsilon_t^2 \alpha)$-RDP, for all $\alpha \geq 1$. Thus, a Rényi filter with parameters $(\alpha, \alpha B^\star)$ would stop once $\frac{1}{2}\sum_{t=1}^k \epsilon_t^2 > B^\star$. Since this condition is independent of $\alpha$, the output of adaptive composition with this stopping condition satisfies $(\alpha, B^\star)$-RDP for all $\alpha \geq 1$. This guarantee is equivalent to $B^\star$-zCDP, and by assumption this implies $(\epsilon_g, \delta_g)$-DP as well.

By Fact 2.4, $B^\star$-zCDP implies $\left( \min_\alpha \alpha B^\star + \frac{\log(1/\delta_g)}{\alpha-1}, \delta_g \right)$-DP. Optimizing over $\alpha$ and solving for $B^\star$ such that $\min_\alpha \alpha B^\star + \frac{\log(1/\delta_g)}{\alpha-1} = \epsilon_g$ yields $B^\star = \left( -\sqrt{\log(1/\delta_g)} + \sqrt{\log(1/\delta_g) + \epsilon_g} \right)^2$.

## A.6   Proof of Proposition 5.1

By properties of the Gaussian mechanism, the individual RDP parameters of order $\alpha$ are $\rho_t^{(i)} = \frac{\alpha \|\bar{g}_t(X_i)\|_2^2}{2\sigma^2}$. Therefore, by properties of the individual filter, as long as $\frac{\alpha \sum_{j=1}^t \|\bar{g}_j(X_i)\|_2^2}{2\sigma^2} \leq B$, the output is $(\alpha, B)$-individually RDP for $X_i$. The clipping step ensures this inequality holds with $B = \frac{\alpha B_{\text{norm}}}{2\sigma^2}$ for all $t \in \mathbb{N}$ and for all data points $X_i$, and therefore the algorithm is $\left( \alpha, \frac{\alpha B_{\text{norm}}}{2\sigma^2} \right)$-RDP.

# B   Tracking privacy loss via multiple filters

A privacy filter is meant to shape the course of adaptive composition by limiting the incurred privacy loss. In practice, one might want to track the privacy loss incurred so far without constraining the

analyses. Rogers et al. [29] formalize this desideratum in terms of a *privacy odometer*. We provide an alternative approach to this task, by observing that a valid Rényi privacy filter can be utilized to design an approximate tracker of the privacy loss.

We denote by $\rho_t$ the RDP parameter of order $\alpha$ of $\mathcal{A}_t$ conditional on the past reports, as in equation (1).

---

**Algorithm 4** Tracking privacy loss via multiple privacy filters

---

**input :** dataset $S \in \mathcal{X}^n$, discretization error $\Delta > 0$, sequence of algorithms $(\mathcal{A}_t)_{t=1}^k$
  Initialize tracker $O_1 = \Delta$, set $T_{\text{restart}} = 1$
  **for** $t = 1, 2, \ldots, k$ **do**
     Compute $a_t = \mathcal{A}_t(a_1, \ldots, a_{t-1}, S)$, set $O_t \leftarrow O_{t-1}$
     **if** $\mathcal{F}_{\alpha,\Delta}(\rho_{T_{\text{restart}}}, \ldots, \rho_t) = \text{HALT}$ **then**
        Augment tracker $O_t \leftarrow O_t + \Delta$, set $T_{\text{restart}} \leftarrow t$
     **end**
  **end**

---

In words, every time an RDP filter with privacy budget $\Delta$ halts, we restart a new filter and augment the tracker by $\Delta$. Here, $\Delta > 0$ is the discretization error of the tracker. An important question here is how one should go about choosing $\Delta$. If $\Delta$ is large, then the tracker is very coarse and inaccurate. On the other end, if $\Delta$ is small, the filters might halt often, and whenever a filter halts we effectively make the upper bound on the tracker a bit looser. Roughly speaking, if we restart at time $t$ we lose a factor of $\Delta - \sum_{j=T_{\text{restart}}}^{t-1} \rho_j$, where $T_{\text{restart}}$ is the last restart time before $t$.

We state the guarantees of Algorithm 4. For all $j \in \mathbb{N}$, let $T_j$ denote the step before the $j$-th time a filter restarts in Algorithm 4. More formally, we can define the sequence $\{T_j\}_j$ recursively as[2] $T_j = \min \left\{ k, \min \left\{ t > T_{j-1} : \mathcal{F}_{\alpha,\Delta}(\rho_{T_{j-1}+1}, \ldots, \rho_{t+1}) = \text{HALT} \right\} \right\}$, where $T_0 = 0$.

**Proposition B.1.** *Fix $m \in \mathbb{N}$, and suppose that $\rho_j \le \Delta$ almost surely, for all $j \in \mathbb{N}$. Then, the tracker in Algorithm 4 satisfies* $\sup_{(S,S') \in \mathcal{S}} D_\alpha^\leftrightarrow \left( \mathcal{A}^{(T_m)}(S) \| \mathcal{A}^{(T_m)}(S') \right) \le m\Delta = O_{T_m}$.

*Proof.* The algorithm $\mathcal{A}^{(T_m)}$ can be written as an adaptive composition of $m$ algorithms, each of which outputs $(a_{T_{j-1}+1}, \ldots, a_{T_j})$, $j \in \{1, \ldots, m\}$. Therefore, by the standard adaptive composition theorem for RDP, it suffices to argue that each of these $m$ algorithms is RDP, conditional on the outputs of the previous algorithms. Since $\mathcal{F}_{\alpha,\Delta}$ is a valid Rényi privacy filter by Theorem 4.3, each of these $m$ algorithms is indeed $(\alpha, \Delta)$-RDP, which completes the proof. □

Notice that Proposition B.1 immediately implies that Algorithm 4 is also valid for any $T$ such that $T_{m-1} \le T \le T_m$, since $(a_1, \ldots, a_T)$ is a post-processing of $(a_1, \ldots, a_{T_m})$.

Proposition B.1 allows designing a personalized tracker $O_t^{(i)}$ for all $X_i \in S$. The update is analogous to that of Algorithm 4, the difference being that a separate filter is applied to the individual privacy parameters of each point. Naturally, each point has its own times of filter exceedances, $\{T_j^{(i)}\}_j$. Notice that the values $O_t^{(i)}$ are *sensitive*, as they depend on the value of $X_i$. Importantly, they can be disclosed to the respective user without violating the other users' privacy; $O_t^{(i)}$ depends on $X_i$, but it does not depend on the other data points (other than through $a^{(t-1)}$, which is reported privately).

## C  Experimental details

We train a convolutional neural network using the implementation of private gradient descent from the Opacus PyTorch library [33]. We use the same architecture as in the MNIST example of the library. Since we run batch gradient descent and not SGD as in the library example, we tune all hyperparameters from scratch. Each step of gradient descent (which is computed using the whole dataset) took about 22 seconds on average on a Macbook Pro from 2019.

---

[2]We think of the minimum of an empty set as $\infty$.

For $\epsilon = 0.3$, we set $\sigma = 170$, $C = 10$, $\eta_t \equiv \eta = 0.2$, and $k = 112$ for private GD without filtering. To achieve the same privacy guarantees using private GD with individual filtering, we set $B_{\mathrm{norm}} = kC^2 = 11200$.

For $\epsilon = 0.5$, we set $\sigma = 130$, $C = 15$, $\eta_t \equiv \eta = 0.15$, and $k = 180$ for private GD without filtering. For GD with individual filtering, we set $B_{\mathrm{norm}} = kC^2 = 40500$.

For $\epsilon = 1.0$, we set $\sigma = 100$, $C = 10$, $\eta_t \equiv \eta = 0.2$, and $k = 420$ for private GD without filtering. This parameter configuration achieves accuracy of $(96.25 \pm 0.23)\%$. When private GD achieves such high accuracies, we observe little benefit to individual filtering. This is due to the fact that the proportion of points filtered out right after round $\lfloor B_{\mathrm{norm}}/C^2 \rfloor$ is comparable to the proportion of points yet misclassified, suggesting that few misclassified points remain in the active pool. Therefore, we set $B_{\mathrm{norm}} = kC^2 = 42000$, and $k_{\max} = k$.

To apply the individual filter, after $kC^2$ steps we continue running gradient descent while adaptively dropping points when their privacy budget is exhausted. In particular, starting with step $\lfloor kC^2 \rfloor$, we query the training accuracy 8 times in intervals of 5 steps (hence, the total number of additional steps is 35). As the final model we take the iterate when the queried training accuracy is highest.

Note that, technically, these additional queries should be reported in a privacy-preserving manner to formally ensure DP. However, these are simple linear queries that can be reported with high accuracy at little additional privacy cost. For $\epsilon \in \{0.3, 0.5\}$, it suffices to report the training accuracy with $1\%$ resolution. Eight such reports require a smaller privacy cost than, say, one or two additional optimization steps. For $\epsilon = 1.0$ it suffices to report the accuracy with $0.1\%$ resolution since the accuracy improvements after $kC^2$ steps are generally smaller for large $\epsilon$. The additional reports for $\epsilon = 1.0$ have the cost of a few dozen extra steps. In either case, the privacy cost of the additional reports is less than $1\%$ of the intended privacy parameter.

In the suboptimal regime, we increase $C$ by a factor of 1.5 for $\epsilon \in \{0.3, 0.5\}$ and accordingly decrease $\sigma$ by the same factor. The number of steps is similarly decreased by a factor of $1.5^2$. We do a similar adjustment for $\epsilon = 1.0$, where we increase $C$ by a factor of 2. These results are shown in the main body of the paper.

Here, we perform similar experiments where we decrease $\sigma$ by a factor of 1.5 for $\epsilon \in \{0.3, 0.5\}$ and by a factor of 2 for $\epsilon = 1.0$. We adjust the number of optimization steps accordingly, and keep all other hyperparameters as in the tuned regime. The results are shown below. As before, we observe benefits to performing additional steps, especially for small values of $\epsilon$.

| $\epsilon$ | GD (suboptimal noise scaling) | GD (suboptimal noise scaling) w/ filtering |
|------|------|------|
| 0.3 | $(86.88 \pm 2.28)\%$ | $(91.20 \pm 0.73)\%$ |
| 0.5 | $(92.37 \pm 1.32)\%$ | $(93.86 \pm 0.39)\%$ |
| 1.0 | $(94.35 \pm 0.23)\%$ | $(94.50 \pm 0.14)\%$ |