# OpenReview forum: "Individual Privacy Accounting via a Rényi Filter"
_NeurIPS.cc/2021/Conference — NeurIPS 2021 Poster_

### Official Review · Reviewer_Aso5 · 2021-06-25

**Rating:** 6
**Confidence:** 4

**Summary:**

This work proposes to track each individual's privacy loss when applying mechanisms to a database to preserve Renyi DP, instead of using the worst-case privacy budget as in the common composition.
More experiments and discussions are needed to support claims in the abstract.

**Limitations And Societal Impact:**

addressed

**Main Review:**

Strengths:
- The idea of giving a tighter bound for privacy by considering each individual is appealing at the first glance
- The extension to the adaptive case is natural, and the algorithm proposed has a simple form in the linear response situation

Weakness:
- The work aims to allow more responses with the same privacy guarantee since the previous bound is conservative. However, the experiment results are not strongly supportive of its effectiveness. More experimental evidence is definitely needed.
- How to deal with the batch update? Is it worth exploiting the individual privacy loss?
- It would be interesting to compare it with the local differential privacy
- Minor problems with the notation and concepts, say, what's the point of Definition 2.5? Also, the Algo. 3 uses a different mechanism as before.

**Time Spent Reviewing:**

1 hour

---

> ### Author Response · Authors · 2021-08-10
> **Response to Reviewer Aso5**
>
> Thank you for your review.
>
> We would be happy to hear suggestions for additional experimental evidence. Note that in the Supplementary material we provide an additional setting where private gradient descent with filtering outperforms standard private gradient descent. In addition, after the submission deadline we realized that we had made a miscalculation in averaging the accuracies in two settings in the table on page 9. Specifically, for $\epsilon = 0.5$, GD (suboptimal) achieves accuracy (92.07 ± 2.07)%, and GD(suboptimal) w/ filtering achieves (94.30 ± 0.58)%, and for $\epsilon=1.0$, GD (suboptimal) achieves accuracy (94.45 ± 0.45)%, and GD(suboptimal) w/ filtering achieves (95.33 ± 0.34)%.
>
> Could you clarify the second question (“How to deal with the batch update? Is it worth exploiting the individual privacy loss?”)? If it is about the computational aspects of our method, please see the detailed response to Reviewer Qurk. In short, our method has a minor computational overhead relative to standard private gradient descent. SGD is more computationally efficient than our algorithm, but it has a significantly worse privacy-utility tradeoff.
>
> We are not sure what you mean by comparing with local differential privacy. Our filter can be applied in the local differential privacy model without any modification (i.e. there is no additional cost to implementing it locally). Please see Remark 4.6.
>
> Definition 2.5 is central to our analysis. We compute individual RDP parameters, defined in Definition 2.5, to implement individual privacy filtering (Algorithm 2). We do not understand what you mean by “Algo. 3 uses a different mechanism as before.” Which mechanism from before are you referring to? Algorithm 3 is a special case of our general individual filtering algorithm, given in Algorithm 2.
>
> We look forward to your clarifications, and please let us know if you have any further questions.

---

### Official Review · Reviewer_7fdb · 2021-07-15

**Rating:** 6
**Confidence:** 4

**Summary:**

This paper present a fully adaptive composition theorem for the Renyi differential privacy that bounds the overall privacy loss by the personalized privacy losses of all data points. The proposed adaptive composition theorem generalizes the vanilla composition theorem for Renyi differential privacy. The individual privacy accounting is designed as a privacy filter, which monitors the individual privacy loss across all individuals and time to make sure that the data points are dropped when cumulative privacy loss is about to violate the privacy budgets. Besides the theoretical guarantees, the training procedure via DP-stochastic gradient descent is also discussed on how to incorporate individual privacy filtering. Empirical study on MNIST dataset also shows improvements on the proposed privacy filtering method.

**Ethical Concerns:**

No ethical concerns is presented in this paper.

**Limitations And Societal Impact:**

Personalized DP is a very important and practical topic in designing a DP mechanism, since the vanilla DP guarantee is often too pessimistic. There is not potential negative societal impact presented in this work.

**Main Review:**

1. Vanilla accounting approaches are too conservative and those harm the accuracy (utility); therefore, the proposed individual accounting approach aims to improve this shortcoming. In Section 4.2, the authors provide a scenario on high-dimensional linear queries where individual accounting leads to better privacy-utility tradeoff. The key is to compare the privacy budget in the worst-case RDP and that in the individual accounting. In particular, when the max norm of $q_t$ is larger than $\sum_{j=1}^t\|q_j(X_i)\|_2^2$ (i.e., each user’s responses are sparse). In high-dimensional setting, it is possible. However, a query function is not usually high-dimensional and this scenario does not seem to be practical. Is there other more practical scenarios where the advantages of individual accounting can be theoretically proven? What are the general scenarios where the individual accounting will outperform vanilla accounting?

2. When the learning model is carefully tuned, the privacy budgets are all spent and therefore in the empirical study on MNIST in Section 5.1, the improvement with filtering is very marginal comparing to GD (tuned). In most machine learning settings, the neural networks are perfectly tuned. The empirical study did not show the cases when it is hard to perfectly tuned the network parameters, e.g., distributed learning, transfer learning. To me,  the experiments are not sufficient to support the claims made in the Introduction.

3. The paper uses a lot of space on Renyi-DP, but there is no empirical study on RDP in the experiments. How does different $\alpha$  in RDP affects the performance of the filtering?

4. The authors emphasize that GD is more computationally efficient comparing to private SGD and batch GD. A more thorough discussion is needed and run time reported.

5. It will be great to show the decision made by the Renyi filter in the experiments, e.g., when to continue and when to halt.

6. When the network weights are suboptimal, the Renyi filter improve the performance. Are the queries sparse in this case?

**Time Spent Reviewing:**

3

---

> ### Author Response · Authors · 2021-08-10
> **Response to Reviewer 7fdb**
>
> Thank you for your time and effort spent on reviewing our paper, as well as the suggestions made in the review. In what follows we respond to your questions.
>
> 1. Note that Corollary 4.7 is not specific to the high-dimensional case; it implies tighter accounting via the individual privacy filter also when $d=1$. Still, we are not sure what you mean by high-dimensional queries not being practical. For example, the queries computed by private gradient descent are indeed high-dimensional linear queries (as discussed in Section 5), and this is an important practical application. Depending on the dataset and parameters of private gradient descent (such as the clipping and noise level), these queries can exhibit different levels of sparsity, and the sparser the queries are the bigger the gains of the individual privacy filter are. We provide evidence of this in Section 5.
> 2. As we have mentioned, there are many important practical settings in which it is either  impossible or very expensive to tune hyperparameters. For example, federated learning or interactive query answering. Our experiments in Section 5 are meant to give a sense of the performance of filtering in slightly suboptimal parameter settings. A more realistic simulation of such a setting appears to be rather difficult and is beyond the scope of our primarily theoretical work. At the same time we welcome suggestions of more realistic experiments.
> 3. Our experiments use the Gaussian mechanism, which ensures RDP for all orders $\alpha$. Hence, the practical behavior of our filter does not change for different levels $\alpha$ for this mechanism. Different $\alpha$ only affect the conversion from RDP to DP (which is ultimately the privacy criterion we aim to achieve), and we simply pick the optimal $\alpha$ for this conversion. Please let us know if we misunderstood your question.
> 4. We do not claim that GD is more computationally efficient than SGD; on the contrary, we explain that SGD is more computationally efficient, but that GD achieves a better privacy-utility tradeoff. Please see the response to Reviewer Qurk for further details. We will report the runtime. Note that, as we say in the Checklist, our experiments were run on a modern laptop in a matter of hours.
> 5. We can add a plot of the distribution of the filtering decisions for all data points, for several different time points over the course of model training.
> 6. Yes, when the parameters of private gradient descent, namely the clipping value and the noise level, are suboptimally set, then the gradient queries are sufficiently sparse and our filter leads to improved performance.
>
> Please let us know if you have any further questions.

---

### Official Review · Reviewer_6u2m · 2021-07-16

**Rating:** 7
**Confidence:** 3

**Summary:**


      The paper proposes an individual privacy accounting via a Renyi filter. In their algorithm, each data point maintains a fixed privacy budget; The privacy filter will adaptively drop points once their personalized privacy loss exceeds the budget. The authors provide a tighter analysis of privacy loss composition by tracking the associated divergences at an individual level.



**Limitations And Societal Impact:**

The authors have addressed the potential negative societal impact.

**Main Review:**

Weak points and question:
I have a question about the experiments. When the epsilon increases, the gain of using the filter decreases. I am wondering if the benefits of the privacy filter will decrease in the low-privacy regime?

Strengths:

1. Their Renyi DP-based individual privacy filter is of huge potential usefulness to the Federated learning community. For example, their private gradient descent example can be generalized to the federated setting where each client can maintain a privacy budget.

2. They provide a new adaptive composition theorem for RDP, which allows a sequence of adaptively chosen privacy parameters during the training.

3. The related work section is helpful. The authors thoroughly compared their work to other personalized DP/data-dependent DP.

**Time Spent Reviewing:**

5.5 hours

---

> ### Author Response · Authors · 2021-08-10
> **Response to Reviewer 6u2m**
>
> Thank you for your time and effort spent on reviewing our paper, and thank you for the question. Yes, as our experiments indicate, the gains of individual filtering decrease as epsilon increases. In general, as the achieved accuracies with standard private gradient descent increase, making any further accuracy improvements becomes increasingly difficult, as the misclassified points become increasingly difficult as well. For this reason, the “leftover” privacy budget that our filter utilizes has a more significant effect in high-privacy regimes. In those regimes there are still reasonably “typical” points that get misclassified, so with a bit of extra privacy budget such points can be learned.
>
> Please let us know if you have any further questions.

---

### Official Review · Reviewer_Qurk · 2021-07-16

**Rating:** 6
**Confidence:** 5

**Summary:**

The paper carries out the “take suprema later” idea in DP. In particular, by moving the supremum over previous algorithm outputs and neighboring datasets outside of the expectation, they design a new filter for RDP that can potentially achieve savings in the DP parameters. As concrete evidences, they provide a theoretical result where a significant saving can be exploited from the sparsity, and a more practical one using MNIST showing that the saving is significant for suboptimal hyperparameters.

**Limitations And Societal Impact:**

The authors have adequately addressed the limitations and potential negative societal impact of their work.

**Main Review:**

This paper carries out the “take suprema later” idea clealy. Their results are stronger and clearer than those in Rogers et al. I have many compliments regarding this in a previous review, so I will not repeat.

Compared to the previous version, the formal statement about sparsity is satisfying. This resolves one of the issues I posed. However, I am not fully convinced about the following issue.

The current presentation of Algorithm 3 doesn't reveal its computational cost. In particular, an expensive loop is presented using universal quantifier and summation. Although the authors remarks that SGD and minibatch does not save time with filtering, it would be easier for readers to see if it is revealed in the algorithm.

On the same issue, I still don't see the running time even in the appendix. If the experiment is run on very expensive machines that only big companies can afford and it takes significantly longer time than SGD, then the authors argument about "suboptimal hyperparameters are more common" is less convincing --- the computational resources can be spent two ways

1. Tune the hyperparameters really hard using SGD
2. Change SGD to the more expensive GD and put all the hope on the filter proposed in this paper

It requires further evidence that 1 is better than 2. In particular, I don't know why there are less experimental details in this version (appendix included) than the previous one.

Overall, I think it is a good paper. However, I'm very eager to hear the authors response. I'd love to change my score if I misunderstood.

**Time Spent Reviewing:**

1.5 this time. A lot last time

---

> ### Author Response · Authors · 2021-08-10
> **Response to Reviewer Qurk**
>
> Thank you for all your time and effort; your suggestions have helped us improve the paper. In what follows we can hopefully clear up your lingering concerns, as we feel there might be a misunderstanding.
>
> Regarding the computational cost of Algorithm 3 (private gradient descent with filtering), the computational overhead of using filtering is very mild. In particular, in the actual implementation of the algorithm, we do not compute the sum of squared gradient norms from scratch at every step $t$. Rather, for every data point we track a single real number corresponding to the running sum of the squared gradient norms, $\sum_{j=1}^{t-1} || \bar g_j(X_i) ||^2$. After step $t$, we merely augment this value by $||\bar g_{t}(X_i)||_2^2$. The universal quantifier that refers to clipping all points individually is not specific to individual filtering. That is, in the usual implementation of private gradient descent each point gets clipped to value $C$, while with filtering it gets clipped to $\min(C, \sqrt{\text{“budget”} - \text{“running sum”}})$. Therefore, the only real computational overhead is the update of the running sum of the squared gradient norms, which is one simple addition per data point. We will clarify in the surrounding text that in the implementation one does not need to recompute the whole sum of squared gradient norms at every step. We will also explicitly state the computational overhead relative to private gradient descent without filtering, as described above.
>
> In the mandatory checklist following the references (page 12) we state that our experiments were run on a modern laptop; they do not require expensive infrastructure. Specifically, each step of gradient descent (which is computed using the whole dataset) took about 22 seconds on average on a Macbook Pro from 2019. We will clarify the runtime in the experimental section.
>
> That said, we think there might be a misunderstanding based on the two options you contrasted (“tune the hyperparameters really hard using SGD” vs “change SGD to the more expensive GD and put all the hope on the filter proposed in this paper”). It is true that SGD is computationally more efficient than GD with filtering, but it is not true that perfectly tuned SGD achieves accuracies in the same ballpark. In our experiments we compare our algorithm to full batch GD without filtering (not SGD). As explained in the second paragraph of this response, those two algorithms have pretty much the same computational cost. On the other hand, SGD is much more computationally efficient than full batch GD (with or without filtering), but it achieves much lower accuracies: for instance, in the Opacus library documentation one can find that a highly tuned version of SGD for $\epsilon = 1.16$ achieves average accuracy of 94.63%, which is roughly the same accuracy we achieve with tuned batch GD and $\epsilon = 0.5$ (see page 9 in the submission).
>
> In short, GD with or without filtering has a much better privacy-utility tradeoff and is more computationally expensive than SGD. GD with filtering has a privacy-utility tradeoff that is never worse than that of GD without filtering, and the computational overhead of implementing filtering is unnoticeable.
>
> Please let us know if you have any further questions. We hope that this explanation clears up your concerns about the computational aspects of our algorithm.

---

> > ### Comment · Reviewer_Qurk · 2021-08-17
> > **DP-GD achieves better accuracy than DP-SGD on MNIST???**
> >
> > Thanks for providing extra information, but I have to say that your response on Algorithm 3 completely missed the point. I made it very clear that I wanted a comparison between Algorithm 3 and DP-**SGD**. However,the word "overhead" in this paragraph always refers to the overhead of Algorithm 3 compared to DP-GD.
> >
> > However, since the device and the epoch time are provided, it indeed answers my question. If the experiments are carried out correctly, this is enough evidence that filtering is useful in practice. But this raises another question: as the authors point out in the response, the table of page 9 seems to suggest that
> > >On MNIST, DP-GD achieves better accuracy than DP-SGD given the same privacy budget.
> >
> > This is the first time I see a message like this and I'm a bit skeptical about it. Is there any previous work or is it your original discovery? In the (very) recent papers [[paper #1]](https://arxiv.org/abs/2007.14191) and [[paper #2]](https://arxiv.org/abs/2103.00039), both from Google, the authors seem to still think DP-SGD achieves the state-of-the-art for MNIST. The quoted message can potentially neutralize part of their main results. If the advantage of DP-GD over DP-SGD generalizes to other datasets, then much of the recent research dedicated to privacy accounting of subsampled mechanisms should be re-directed elsewhere.
> >
> > Overall, I need the authors confirm two things:
> > 1. if the quoted message is what they meant
> > 2. if it comes from any prior work.
> >
> > If the first answer is yes, then I don't think I'm the only one who "misunderstood". The paper should highlight it more than its current form as some main contributors of the field are unaware of it. If the message comes from another paper, I would definitely be more than happy to learn about it, disseminate it and rethink about my own research. The rating will be changed as long as you promise to highlight it more and cite properly. But if it is your original work, then I cannot recommend an accept, because the message is probably at least significant as the filtering itself, and it is not supported by any code or experimental results explicitly.
> >
> > It would also help me a lot if the AC and/or my fellow reviewers would like to make comments.

---

> > > ### Author Response · Authors · 2021-08-20
> > > **DP-GD and DP-SGD**
> > >
> > > Thank you for the continued discussion, and we apologize for any confusion our previous response might have caused. In what follows we hope to clarify our take on DP-SGD vs DP-GD.
> > >
> > > First, we are not aware of any prior work that has tried running differentially private GD when training CNNs. The likely reason for this is that, while this is less of an issue for smaller datasets such as MNIST, standard implementations of DP-SGD run into memory issues when the batch is set to the full dataset on, say, CIFAR and larger datasets. As a result, this setting is typically not included in the hyperparameter search. At the same time, GD enjoys a boost in privacy accounting since privacy-amplification-by-subsampling with fraction B/n is somewhat less efficient than reducing the sensitivity by a factor B/n.
> > >
> > > We do not make a claim as general as the one you highlighted above. What we have observed in our experiments is that, using the *specific* architecture from the Opacus library, tuned batch GD improves over tuned SGD across several different values of $\epsilon$. We do not know how general this phenomenon is. However, existing theory for DP-ERM suggests that full GD is no worse than SGD (e.g. note (2) in BST14 [https://arxiv.org/pdf/1405.7085.pdf]). Furthermore, the newest large-scale experiments [https://arxiv.org/pdf/2108.01624.pdf] (coincidentally also from Google) point to larger batches being better albeit at the expense of higher computational cost. So it is possible that DP-GD is better than (or comparable to) DP-SGD in many other settings. We will clarify and highlight our observations about the relative performance of GD and SGD in the setting in our paper, and we have included all parameter configurations so that our results can be verified.
> > >
> > > Finally, while we understand the interest in our empirical results on MNIST, our contribution is first and foremost a new conceptual and technical understanding of composition in differential privacy. In particular, we significantly simplify and improve on the privacy filters from the NeurIPS 2016 paper (that did not include any experiments). Composition is a most basic tool in privacy analysis and our results can be applied to a variety of problems. The empirical results are meant primarily to show that individual filters we define are practical and easy to implement. We agree that the included experiments are limited in scope but hope that they will be useful for future exploration.

---

> > > > ### Comment · Reviewer_Qurk · 2021-08-22
> > > > **Clear theory, confusing experiments**
> > > >
> > > > Thank you for providing further details. I agree that the major contribution is theory and I appreciate it. It helped me understand Rogers et al NeurIPS 2016 paper as well. The theory part also improves compared to the previous version.
> > > > I'd like to see the theory part published, but unfortunately the experimental part is a minus, not a plus. As the authors pointed out, this phenomenon has never been observed on MNIST. It could be a surprise to many researchers in the community and could change how they conduct their research. I cannot trust such an important message by default from a paper without code or even explicit words mentioning this phenomenon. Therefore, I have reason to be skeptical about the experiments, and hence about the improvement compared to the ICML submission. My conclusion is also that this paper is promising (probably even more promising than before because the phenomenon might be true and impactful) but it needs more work.
> > > >
> > > > The two papers mentioned cannot serve as convincing evidences that your experiments are carried out correctly. BST14 is a good theory paper, but using their results as a heuristic for CNN seems a big leap. The August paper from Google works on a very different setting: super large dataset, super large model and a different network structure. The easiest way is to provide the code.

---

> > > > > ### Author Response · Authors · 2021-08-24
> > > > > **GD only uses the Opacus library**
> > > > >
> > > > > Thank you for your response. While we do not understand the source of skepticism, we will gladly provide additional details and evidence.  *For our experiments with DP-GD we used only the publicly available Opacus library* (specifically, an earlier version thereof). Our results for DP-GD can be reproduced using the library code with absolutely no change, by running mnist.py with parameters as outlined in Section C of our Supplementary material (and setting the sampling rate to 1). For example, in mnist.py one only needs to make the following changes to the following parameters to run the experiments for $\epsilon=1$:
> > > > >
> > > > > "--sample-rate",
> > > > > default=1,
> > > > >
> > > > > "--epochs",
> > > > > default=420,
> > > > >
> > > > >  "--lr",
> > > > > default=0.2,
> > > > >
> > > > >  "--sigma",
> > > > > default=100.0,
> > > > >
> > > > > "--max-per-sample-grad_norm",
> > > > > default=10.0,
> > > > >
> > > > >  The additional results with filtering rely on minimal changes to the Opacus code outlined in our pseudocode, which is why we didn’t include the code. We will include the code in the final version to make verification of all of our results straightforward.
> > > > >
> > > > > We also found that a very recent exploration of differentially private optimization (and the most thorough we are aware of) came to the same conclusion about DP-GD improving on DP-SGD. It appears to be still in the process of being published but was presented at the TPDP workshop at ICML 2021[https://icml.cc/Conferences/2021/ScheduleMultitrack?event=8376] last month. Please check slide 76 of the third contributed presentation on Fri 10:40 a.m. - 11:25 a.m by Ryan McKenna. The title of the slide is “Full batch gradients are best”. The authors of this work did not seem to find this result particularly surprising (and neither do we). We agree that investigating the tradeoffs between DG-GD and DP-SGD is a valuable direction going forward, but this is an independent research project and as such we do not see it as invalidating any strengths of our paper.

---

> > > > > > ### Comment · Reviewer_Qurk · 2021-09-02
> > > > > > **Thank you for providing the details**
> > > > > >
> > > > > > I would like to thank the authors for the protracted discussion and for providing enough details to verify the code. Unfortunately I'm unable to set up the environment for opacus properly in time. Tensorflow/privacy (which is what I'm familiar with) with the designated parameters seems very memory inefficient and much more time consuming than opacus. I apologize for not being able to prove or disprove what I'm curious about.
> > > > > >
> > > > > > Even if I was able to reproduce all the experimental results, I still keep my opinions that the authors should (at least for the tuned part) compare general DP-SGD (including DP-GD as a special case) with filtered GD. If hyperparameter tuning shows that optimal accuracy of DP-SGD is achieved at sampling_rate=1, then this should be explicitly explained and supported by data and code and shouldn't be assumed to be common knowledge as in the current submission.
> > > > > >
> > > > > > I have to say that my initial concern regarding where the computational resource should be spent has been resolved. What I'm still uncomfortable with is how the experiments are presented, which makes the relative improvement from the ICML submission less sustained. I decide to keep my initial score.

---

> > > > > > > ### Author Response · Authors · 2021-09-02
> > > > > > > **Thank you for following up**
> > > > > > >
> > > > > > > Thank you for your continued engagement, we appreciate your time. As we said in a previous response, we will gladly highlight and clarify our observations about SGD and GD (and support these observations with experimental data and code). Our experiments’ main focus was on showing that our conceptual tools and techniques are practical and easy to apply. Our goal was to find an easy-to-implement setting in which one can compare usual accounting against individual accounting. Since individual accounting in the context of gradient descent has virtually the same computational properties as batch gradient descent, that seemed like a natural baseline. The benefit of using full batches instead of SGD, while interesting in its own right, is not due to individual accounting so it is outside the scope of our work (and, conveniently, there is the recently presented work that explores it).

---

### Official Review · Reviewer_3v4m · 2021-07-18

**Rating:** 7
**Confidence:** 4

**Summary:**

Overview: Privacy filters are probabilistic objects which allow one to study the composition of differentially private algorithms in a setting where the privacy parameters can be chosen adaptively. While filters have been constructed for the case of pure and approximate differential privacy, they have not been studied in the context of Renyi privacy. Moreover, there hasn’t been any study of individual privacy filters, objects which selectively drop elements out of the dataset when their privacy loss exceeds a certain preselected threshold.

This paper introduces Renyi privacy filters, objects providing theoretical guarantees for the composition of Renyi DP algorithms where the privacy rates can be chosen adaptively. In addition, the authors extend these filters to the individual level, allowing for selective removal of datapoints from the dataset in order to continue computation without exceeded a predetermined level of privacy. The authors provide a comparison of their privacy filters to others, showing significant closed-form improvement in several cases. Likewise, they provide experiments showing the application of individual privacy filters to private gradient descent.


**Limitations And Societal Impact:**

The authors do adequately address limitations. For instance, in the experimental section, they discuss that they often times only see limited empirical improvement using individual privacy accounting. Otherwise, the theoretical contributions generally improve on pre-existing results.

**Main Review:**

Originality: While privacy filters are not original to this, the extension of the concept to Renyi DP is useful as many of the tightest analyses of common mechanisms rely on this form of privacy. The mathematical analysis does not leverage any new techniques, but it does provide a clean solution to the problem. Individual privacy accounting is novel and seems like an invaluable way of extending the number of computations which can be done on a dataset.

Quality: The paper is mathematically sound. Concepts are well motivated and rigorously presented. Some more investigation could have been done into using RDP filters to construct filters for approximate differential privacy, as this seems like one of the most promising applications of the results of the paper.

Clarity: The writing is mostly, and the paper flows well. Some sections (such as the experiments or applications to approximate DP) could have been further touched upon, as these really show the value of the theoretical contributions. Privacy loss is key to the paper, so perhaps it should be discussed more in the main body of the paper as opposed to the appendix.

Significance: The paper is the first to discuss Renyi filters, and leverages them to improve upon the original privacy filters found in [29].

---------------------------------------------------------------------------------------------------------------------------------------------------------------------------------------

High-level Positives:
-	Constructing privacy filters for Renyi DP is very useful, as Renyi DP is used in many privacy accountants like those found in Tensorflow and PyTorch.
-	Likewise, the improvement over the original privacy filters in special cases (e.g. pure differential privacy) is impressive, especially given that the obtained bounds are (a) more readable, and (b) more closely match advanced composition.
-	The proofs are all clean and easy to follow, which helps make the presentation of Renyi Filters very clear.


High-level Negatives:
-	The proofs are mostly straightforward and do not introduce novel analysis techniques. The main mathematical arguments just apply standard results from martingale theory or standard conversion results between forms of privacy.
-	The use of Renyi Filters as approximate DP filters is very interesting. More comparisons would be valuable- even numerical ones if getting a closed form expression isn’t possible.

Additional Comments:

Section 1:
-	The problem is very well motivated, and the notions of filters, both at the individual and overall level are clear. Moreover, the relation to existing work is clear as well.


Section 3:
-	It might be useful to formally define the privacy loss in the body of the paper, as it is the key theoretical object used to prove results. Moreover, it is ultimately the object the filters are trying to control.
-	In the statement of Theorem 1, you state that the sequence of algorithms is of finite length, consisting of k algorithms. I think you can have it be of infinite length and still be able to apply the optional stopping Theorem in the proof.

Section 4:
-	You handle the case of pure differential privacy, but only provide an explicit bound for the case where all privacy parameters, e_t, are chosen to be the same in each round. I think it would be very informative to show more general cases to demonstrate superiority over the bounds found in Rogers et. al.
-	Pure privacy filters don’t immediately yield approximate privacy filters. There is a conversion cost necessary that can be quite large depending on the choice of privacy level epsilon. This wasn’t in the original version of [29], but more recent version have addressed this gap as an error was pointed out (such as the newest version on Arxiv I believe).

Section 5:
-	The experiments are interesting. Due to the fact the privacy filters allow you to change privacy levels based on previous algorithm outputs, it would be interesting to see if varying the privacy level as one goes would affect the convergence rate of gradient descent.


**Time Spent Reviewing:**

4

---

> ### Author Response · Authors · 2021-08-10
> **Response to Reviewer 3v4m**
>
> Thank you for your exceptionally thoughtful and thorough review, as well as for helpful suggestions. We will incorporate a discussion of the privacy loss and an extended comparison to the filter of Rogers et al. in the main body.
>
> We just realized that a corrected version of the Rogers et al. paper appeared on arxiv yesterday, and we have not yet had a chance to look at the updated version before posting this response. We will read through the revised version of the paper in the coming days and update our paper accordingly.
>
> Please let us know if you have any questions and we would be happy to clarify.

---

### Decision · Program_Chairs · 2021-09-27

**Decision:**

Accept (Poster)

**Comment:**

This paper provides new results for (differential) privacy filters that ensure that the individual privacy loss does not exceed certain thresholds. This is an important question in differentially private machine learning, and the paper gives non-trivial technical contributions. During the discussion, reviewers raised some questions regarding the experiments. The authors should provide further details about their experimental setup in the next revision.